# Controls on surface water carbonate chemistry along North American ocean margins

Wei-Jun Cai [1✉], Yuan-Yuan Xu [1], Richard A. Feely[2], Rik Wanninkhof[3], Bror Jönsson [4], Simone R. Alin [2], Leticia Barbero[3,5], Jessica N. Cross[2], Kumiko Azetsu-Scott[6], Andrea J. Fassbender [7], Brendan R. Carter[2,8], Li-Qing Jiang[9], Pierre Pepin[10], Baoshan Chen [1], Najid Hussain[1], Janet J. Reimer[1], Liang Xue [11], Joseph E. Salisbury[12], José Martín Hernández-Ayón[13], Chris Langdon [14], Qian Li [1], Adrienne J. Sutton [2], Chen-Tung A. Chen [15] & Dwight K. Gledhill[16]

Syntheses of carbonate chemistry spatial patterns are important for predicting ocean acidification impacts, but are lacking in coastal oceans. Here, we show that along the North American Atlantic and Gulf coasts the meridional distributions of dissolved inorganic carbon (DIC) and carbonate mineral saturation state ($\Omega$) are controlled by partial equilibrium with the atmosphere resulting in relatively low DIC and high $\Omega$ in warm southern waters and the opposite in cold northern waters. However, pH and the partial pressure of $CO_2$ ($pCO_2$) do not exhibit a simple spatial pattern and are controlled by local physical and net biological processes which impede equilibrium with the atmosphere. Along the Pacific coast, upwelling brings subsurface waters with low $\Omega$ and pH to the surface where net biological production works to raise their values. Different temperature sensitivities of carbonate properties and different timescales of influencing processes lead to contrasting property distributions within and among margins.

[1] School of Marine Science and Policy, College of Earth, Ocean, and Environment, University of Delaware, 111 Robinson Hall, Newark, DE 19716, USA. [2] NOAA Pacific Marine Environmental Laboratory, 7600 Sand Point Way NE, Seattle, WA 98115, USA. [3] NOAA Atlantic Oceanographic and Meteorological Laboratory, 4301 Rickenbacker Causeway, Miami, FL 33149, USA. [4] Plymouth Marine Laboratory, Prospect Place, Plymouth PL1 3DH, UK. [5] Cooperative Institute for Marine & Atmospheric Studies, University of Miami, 4600 Rickenbacker Causeway, Miami, FL 33149, USA. [6] Fisheries and Oceans Canada, Bedford Institute of Oceanography, 1 Challenger Drive, Dartmouth, NS B2Y 4A2, Canada. [7] Monterey Bay Aquarium Research Institute, 7700 Sandholdt Road, Moss Landing, CA 95039, USA. [8] Joint Institute for the Study of the Atmosphere and Ocean, University of Washington, 3737 Brooklyn Avenue NE, Seattle, WA 98195, USA. [9] Earth System Science Interdisciplinary Center, University of Maryland, College Park, MD 20740, USA. [10] Department of Fisheries and Oceans, Northwest Atlantic Fisheries Centre, St. John's, NL, Canada. [11] First Institute of Oceanography, Ministry of Natural Resources, 266061 Qingdao, China. [12] Ocean Process Analysis Laboratory, University of New Hampshire, Durham, NH 03824, USA. [13] Instituto de Investigaciones Oceanológicas, Universidad Autónoma de Baja California, Ensenada, Baja California, Mexico. [14] Department of Marine Biology and Ecology, University of Miami, 4600 Rickenbacker Causeway, Miami, FL 33149, USA. [15] Department of Oceanography, National Sun Yat-sen University, Kaohsiung 80424, Taiwan, ROC. [16] NOAA Ocean Acidification Program, 1315 East-West Highway, Silver Spring, MD 20910, USA. ✉email: wcai@udel.edu

A bsorption of anthropogenic $CO_2$ from the atmosphere has acidified the ocean, as indicated by increases in sea surface $pCO_2$ and hydrogen ion concentration ($[H^+]$) and decreases in pH, carbonate ion concentration ($[CO_3^{2-}]$), and calcium carbonate ($CaCO_3$) mineral saturation state[1–5]. The latter is a ratio of the ionic product, $[Ca^{2+}][CO_3^{2-}]$, to its saturated value below which $CaCO_3$ dissolution may occur (in this paper we only target the saturation state of aragonite, $\Omega_{arag}$, a mineral form that comprises the shells and hard parts of corals, mollusks, and many other marine organisms). More than a decade of ocean acidification research has improved our understanding of the mechanisms controlling the global spatial patterns and temporal variations of ocean carbonate chemistry[5–8], and has begun to reveal how ocean chemistry (e.g., pH and $\Omega_{arag}$)[9–13] and marine organisms[14–16] are responding to anthropogenic $CO_2$ uptake. In coastal regions, much of the ocean acidification research has focused on examining how anthropogenic $CO_2$-induced acidification is mitigated or exacerbated by biological production (removing $CO_2$) in surface waters and subsequent respiration (adding $CO_2$) in subsurface waters as a result of natural nutrient enrichment or human-induced coastal eutrophication[17–20]. Attention has also been given to the effects of biological community composition and community metabolism on the potential to buffer or exacerbate ocean acidification[21,22]. There is, however, incomplete knowledge of large-scale patterns of carbonate chemistry and the mechanisms controlling its variability in coastal oceans, largely due to limited observations and the added complication of dynamic regional or local processes that can be large in magnitude.

Examples of local processes that influence surface water distributions of $pCO_2$, pH, total alkalinity (TA), dissolved inorganic carbon (DIC), and $\Omega_{arag}$ in ocean margins include river and wetland inputs, coastal circulation, vertical and lateral mixing, spatial and seasonal temperature variations, the balance of biological production and respiration, and anthropogenic $CO_2$ uptake[23–30]. Rivers usually carry more acidified waters with a higher DIC/TA ratio than seawater, which may weaken the ability of coastal seas to withstand anthropogenic $CO_2$-induced acidification[29,31]. Freshwater input can also suppress $\Omega_{arag}$ by reducing $[Ca^{2+}]$. However, increased nutrient input from rivers may lead to elevated biological $CO_2$ removal or basification in surface waters[32], whereas the respiration of this organic material back to $CO_2$ in bottom waters may enhance acidification at depth[17]. Additionally, coastal upwelling can bring low pH and low $\Omega_{arag}$ waters to shallow, nearshore regions and put coastal biological systems in stress[27]. Subsequent $CO_2$ release to the atmosphere and biological production stimulated by the accompanying upwelled nutrients may increase pH and $\Omega_{arag}$[33].

In addition to these driving processes, it is important to understand the influence of temperature on the spatial distributions of carbonate system properties. Coastal currents often bring waters from warm (or cold) locations to cold (or warm) locations, resulting in chemical equilibrium shifts and air–sea gas exchange, both of which are sensitive to temperature change:

$$CO_{2gas} \xleftrightarrow{K_0} CO_{2aq}, \tag{1}$$

$$CO_{2aq} + H_2O + CO_3^{2-} \xleftrightarrow{K_1/K_2} 2HCO_3^-. \tag{2}$$

Here $K_0$ is the solubility constant, and $K_1/K_2$ is the ratio of the first and second dissociation constants of carbonic acid. When a water mass isolated from the atmosphere (referred to herein as a closed system) is cooled, the ratio, $K_1/K_2$, will increase (Fig. 1a), and the acid–base equilibrium will shift to the right to reduce dissolved $CO_2$ ($CO_{2aq}$) and $CO_3^{2-}$ and increase $HCO_3^-$ (Eq. 2). The gas solubility constant will also increase, further reducing

$pCO_2$ following Henry's Law ($pCO_2 = [CO_{2aq}]/K_0$) (Fig. 1a–c), as gas exchange is not allowed to compensate the cooling process in a closed system (Eq. 1). Although not explicitly shown in Eq. (2), cooling will also reduce $[H^+]$ and increase pH in a closed system because of the decrease in both acid dissociation constants (Fig. 1). Importantly, cooling can shift a water mass from being a source of $CO_2$ to the atmosphere in warm, subtropical waters to a sink for $CO_2$ in cold, mid-latitude waters when gas exchange occurs in an alongshore current. The process of cooling and gas exchange in an open-system increases $CO_{2aq}$, a fraction of which combines with and reduces $CO_3^{2-}$ and thus $\Omega_{arag}$.

To illustrate the temperature dependences of the marine carbonate system, we define two thermodynamic end members: the closed system, with no gas exchange, and the open system, with a complete $CO_2$ gas exchange equilibrium between seawater and the atmosphere. Thus, our theoretical framework posits that the temperature-dependent change of a species or a property in an open system is the combined effect of gas exchange caused by the solubility change and the subsequent internal thermodynamic equilibrium shift in a closed system (Fig. 1). Importantly, these two temperature effects counteract each other for $[CO_{2aq}]$ (hereafter the subscript aq will be omitted), $pCO_2$, $[H^+]$, and pH, whereas they work to enhance each other for $[CO_3^{2-}]$ and $[HCO_3^-]$ (or roughly DIC), with atmospheric equilibration playing the dominant role in an open system (Fig. 1). In summary, the combined result of the thermodynamic equilibrium shift and atmospheric equilibration during cooling is predicted to be an increase in $[CO_2]$ and DIC, essentially no change in $[H^+]$ and pH and, by definition, no change in $pCO_2$, and a relatively large decrease in $[CO_3^{2-}]$ and $\Omega_{arag}$ in surface waters (Fig. 1)[7,8,34–36]. Therefore, spatial variations in pH and $pCO_2$ largely reflect air–sea disequilibrium caused by local physical and biological processes that act more rapidly than gas exchange.

Here we report results from large-scale, regional marine carbonate chemistry observing efforts on the Atlantic, Gulf, and Pacific coasts of North America. This synthesis reveals the domain-scale and local controls on carbonate properties driven by atmospheric $CO_2$ exchange and physical, chemical, and biological ocean processes. Our comparative study of marine carbonate chemistry across physically and biologically dissimilar ocean margins provides new insight about the mechanisms controlling $CO_2$ parameter distributions, timescales of variability, and responses to ocean warming and acidification.

## Results

**The Atlantic and Gulf coasts.** The North American Atlantic and Gulf of Mexico (GOM) coastal regions are characterized by broad, shallow shelves influenced on the landward side by rivers and wetlands and on the seaward side by alongshore currents. Most notably, the northbound Gulf Stream Current system brings warm, high-salinity source waters from the tropics, while the southbound Labrador Current brings cold, low-salinity source waters from the Arctic and subarctic regions (Supplementary Fig. 1)[37–39]. While data presented here were collected during slightly different summer months on the Atlantic coast (Supplementary Table 1), the spatial patterns of all parameters are consistent for the cruises conducted in 2007, 2012, and 2015, each plotted with a shift in longitude for the purpose of illustration (Fig. 2, Supplementary Table 2).

Sea surface temperature (SST) (Fig. 2a) and salinity (SSS) (Fig. 2b) decrease from southwest to northeast. The surface TA distribution follows that of salinity, with the highest TA values in the GOM, a sharp south-to-north decrease along the Atlantic coast[24,25], and nearshore modifications by major rivers[40] (Fig. 2c). In contrast to the sharp meridional decline in TA, the DIC gradient is weak along the Atlantic coast, exhibiting smaller south-

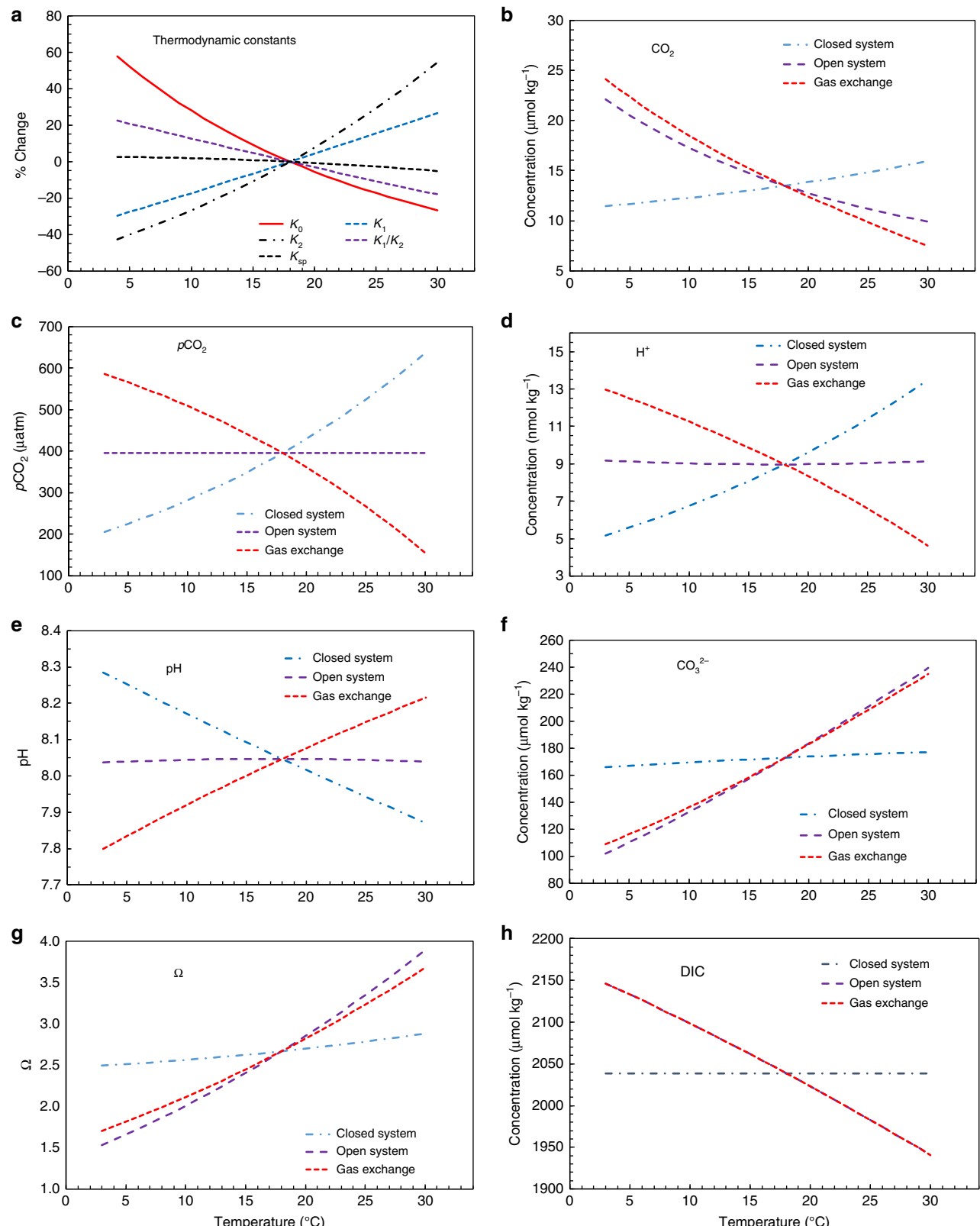

to-north decreases and some high values in the northern regions (Fig. 2d). Thus, the DIC/TA ratio increases sharply from south-to-north (Fig. 2f), which strongly correlates with SST (Supplementary Table 3). The DIC/TA ratio is a measure of the acid–base equilibrium point of seawater, with higher ratio indicating higher $CO_2$ fraction in the DIC pool and more acidified, less buffered

states[41]. Note, while the GOM has both high DIC and TA values, its DIC/TA ratio is the lowest, indicating the least acidified state of all the North American margins. In addition, elevated TA values in North America's largest river, the Mississippi, cause the northern GOM to have one of the highest TA-to-salinity ratios (~75) in the world ocean (Fig. 2e)[29,36,42,43].

**Fig. 1 Temperature dependence of $CO_2$ system species and thermodynamic constants. a** Thermodynamic constants, **b** $CO_2$, **c** $pCO_2$, **d** [$H^+$] (total concentration), **e** pH (on the total proton scale), **f** [$CO_3^{2-}$], **g** $\Omega_{arag}$, and **h** DIC. Here $K_0$ is Henry's Law constant, and, by convention, dissolved $CO_2$ (~99.5%) and $H_2CO_3$ (~0.5%) are not differentiated and are combined as one in [$CO_{2aq}$] (or in [$H_2CO_3$]). The first acid dissociation reaction step is $H_2CO_3 = H^+ + HCO_3^-$ with its equilibrium constant defined as $K_1 = [H^+][HCO_3^-]/[H_2CO_3]$, whereas the second dissociation step is $HCO_3^- = H^+ + CO_3^{2-}$ with its equilibrium constant defined as $K_2 = [H^+][CO_3^{2-}]/[HCO_3^-]$. A combined equilibrium constant $K_1/K_2$ is defined for the combined reaction (Eq. 2). The open system is calculated under an assumption of water in full equilibrium with the atmosphere or at a constant $pCO_2$ condition with $pCO_2 =$ 395 µatm, TA = 2280 µmol kg$^{-1}$, and S = 35. At 18 °C, the open system has a DIC = 2038.6 µmol kg$^{-1}$. The closed system is calculated under an assumption of no $CO_2$ gas exchange with the atmosphere or at a constant DIC condition with TA = 2280 µmol kg$^{-1}$, DIC = 2038.6 µmol kg$^{-1}$, and S = 35. For the purpose of illustrating the carbonate chemistry behavior, and for convenience, we set 18 °C as the reference point where the closed system and the open system have the same parameters. In this case, the gas exchange term = open system (T) − closed system (T) + open system (18 °C), where T is the temperature. Thus, the gas exchange term reflects DIC increases (or decrease) at lower (or higher) temperature by taking up $CO_2$ from (or releasing $CO_2$ to) the atmosphere and the redistribution of DIC among various species via acid–base equilibrium shifts.

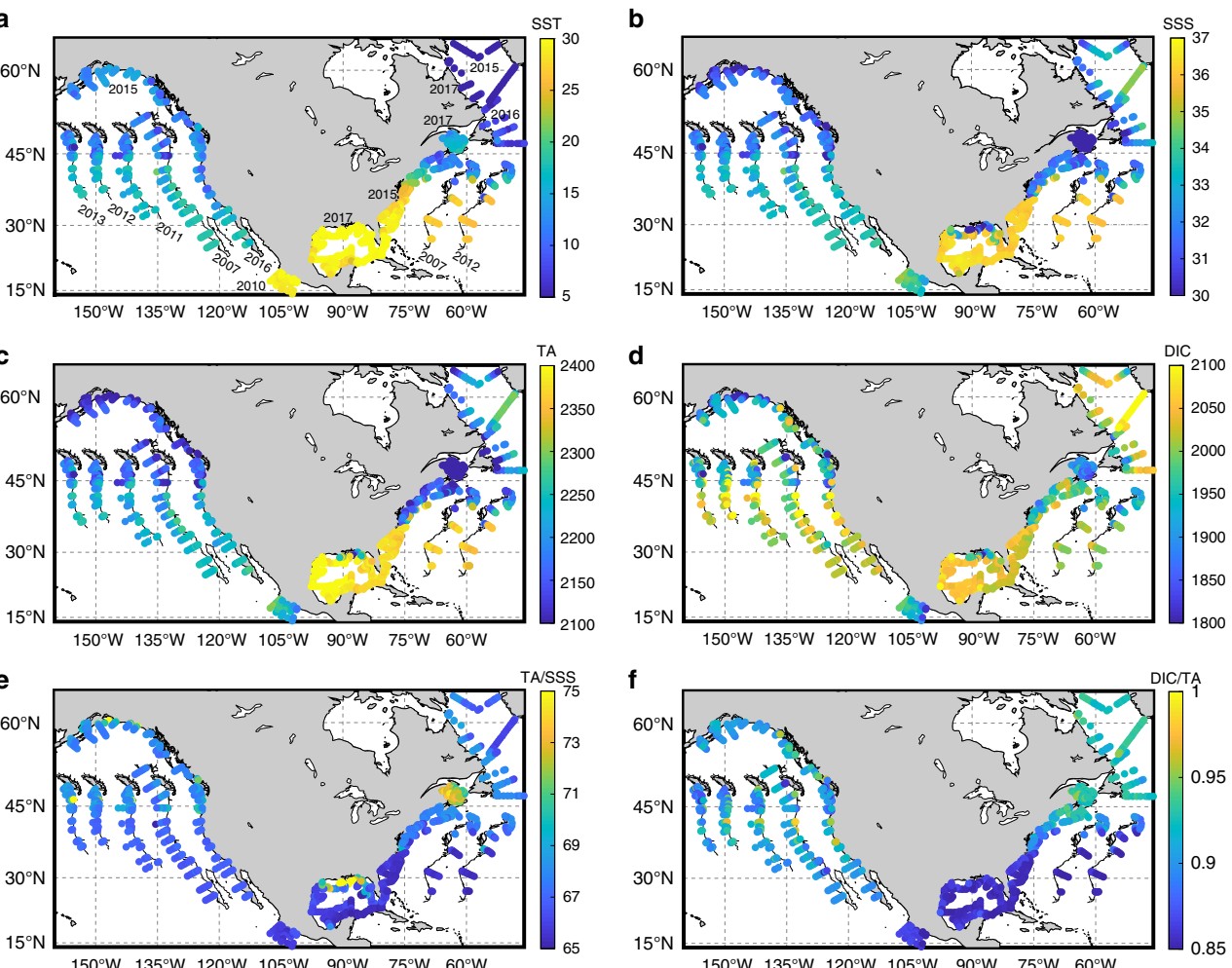

**Fig. 2 Distributions of physical and biogeochemical parameters.** These parameters are important to understand the marine carbonate system in surface seawater along the North American margins. **a** Sea surface temperature (SST), **b** sea surface salinity (SSS), **c** total alkalinity (TA), **d** total dissolved inorganic carbon (DIC), **e** TA/SSS ratio, and **f** DIC/TA ratio. See the "Methods" section for details about the measurement or calculation of these parameters. To conserve space, only the 2015–2017 data are presented in their exact geographic locations on a North American ocean margin map, while earlier data are plotted at the correct latitude with a longitude offset to show that the general patterns presented here are persistent across cruises conducted in different months and years. Cruise dates and general locations are given in the "Methods" section and Supplementary Table 2.

Sea surface $pCO_2$ and pH show more complex spatial patterns (Fig. 3a, c). These parameters exhibit no significant co-variation with SST and weak (or no) co-variation with DIC/TA and the percent saturation of dissolved oxygen (DO%) in surface waters, an indicator of biological activity (Supplementary Fig. 2 and Table 3). This is likely due to the influence of multiple competing processes along the Atlantic and GOM coasts. By normalizing the

$pCO_2$ and pH values to 25 °C (i.e., setting the thermodynamic constants to 25 °C in a closed-system calculation), a south-to-north pattern becomes apparent with high $pCO_{2@25°C}$ and low pH$_{@25°C}$ values in the cold northern regions. A comparison of $pCO_2$ and pH distributions at in situ SST and at 25 °C not only reveals the important role of temperature in seawater carbonate system thermodynamic equilibrium, but, perhaps more

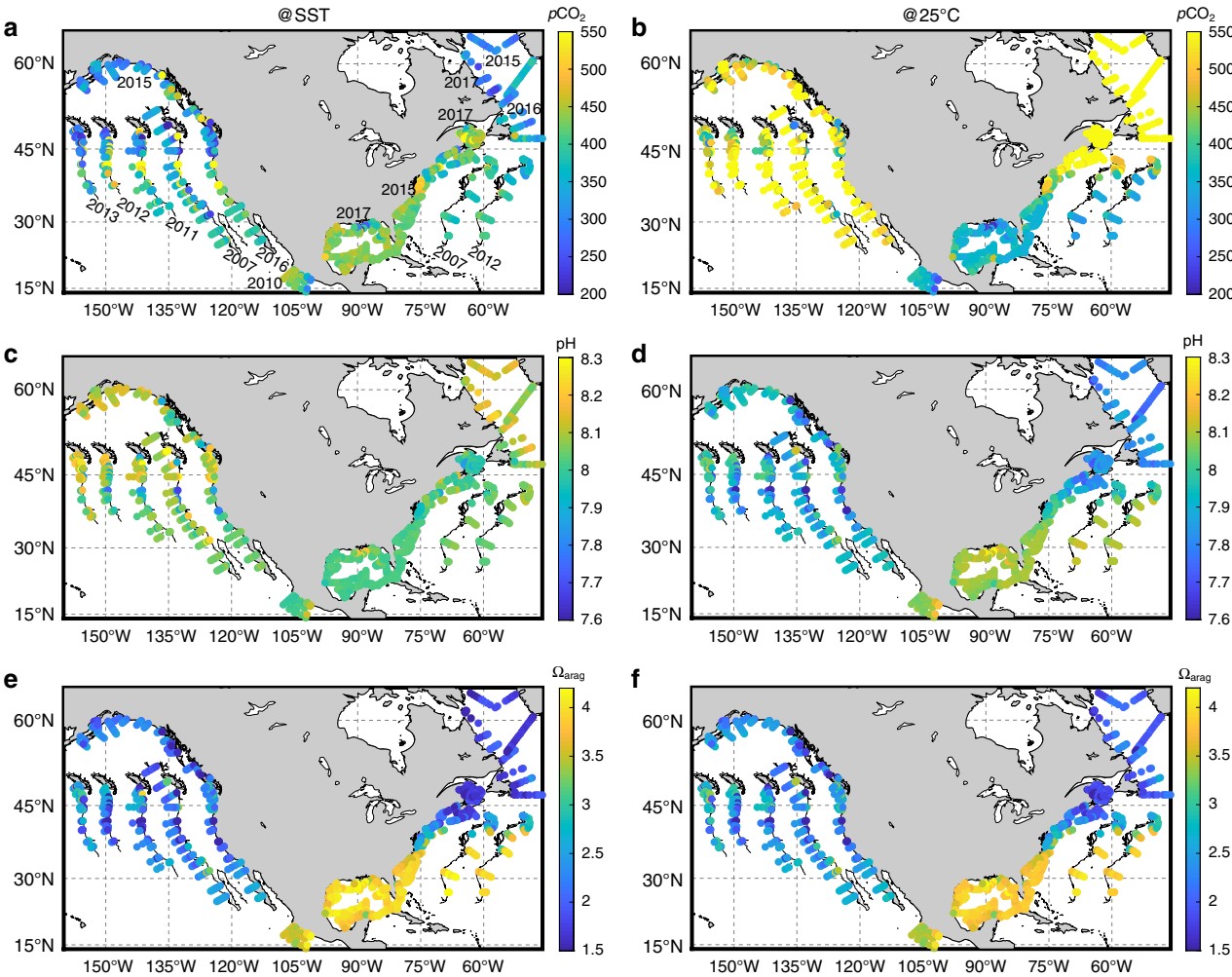

**Fig. 3 Distributions of marine carbonate system parameters.** Shown are parameters in surface seawater along the North American margins at field temperature and normalized to 25 °C. Left column panels are at field temperature (@SST), while the right column panels are at 25 °C. **a**, **b** Partial pressure of $CO_2$ ($pCO_2$), **c**, **d** pH (on the total scale), and **e**, **f** aragonite saturation state ($\Omega_{arag}$). Note that the temperature normalization from field SST to 25 °C is performed under the assumption of a closed system with no gas exchange between surface seawater and the atmosphere (thus no DIC change). As in Fig. 2, only the most recent data are presented in their exact geographic locations, while earlier data are plotted at the correct latitude with a longitude offset.

importantly, also the role of gas exchange in eliminating the temperature-induced air–sea disequilibrium. In other words, high DIC/TA exists in cold northern waters mainly due to atmospheric $CO_2$ uptake induced by low SST. In mid-latitude regions, low observed $pCO_2$ and high observed pH appear in areas of high DO %, suggesting biological $CO_2$ removal there (e.g., early summer in the Gulf of Maine, Fig. 3 and Supplementary Fig. 2)[44].

In northern GOM coastal waters, low $pCO_2$ and high pH values (at both SST and 25 °C, Fig. 3a–d) strongly correlate with the low salinity, high DO%, and low DIC/TA ratios in the nutrient-rich Mississippi River plume-influenced region, revealing the importance of local biological $CO_2$ removal driven by riverine nutrients (Supplementary Table 3). The rest of the GOM has relatively high $pCO_2$ and low pH, reflecting the warm climate and low biological production in offshore waters[42]. Carbonate precipitation is another process that could be contributing to the relatively high $pCO_2$ and low pH conditions. This is supported by a slightly lower TA/SSS ratio (Fig. 2e) and a higher DIC/TA ratio (Fig. 2f, Supplementary Fig. 4) in southern GOM, indicating more TA removal than DIC removal. Precipitation likely occurs in

waters above $CaCO_3$-rich banks at the Florida Keys and Yucatan peninsula, but can also happen in other areas[45].

Most strikingly, along the entire Atlantic and GOM coasts, the large-scale spatial distribution of $\Omega_{arag}$, an important metric for ocean acidification stress on organisms, bears little resemblance to the distributions of $pCO_2$ and pH (Fig. 3). Rather, the $\Omega_{arag}$ distribution is similar to SST and is inversely correlated with the DIC/TA ratio ($r = -0.996$, $p < 0.001$ in the Atlantic, see Supplementary Table 3 for other margins). Thus, similar to the strong DIC/TA and temperature gradients, there is a strong south-to-north decline in $\Omega_{arag}$. Interestingly, $\Omega_{arag}$ and its associated parameters (e.g., $[CO_3^{2-}]$, Supplementary Fig. 3), but not $pCO_2$ and pH, change abruptly at Cape Hatteras, where the Gulf Stream moves offshore and the influence of the Labrador Current increases in coastal waters. The highest $\Omega_{arag}$ appears in the GOM, which is consistent with the existence of the highest SST and TA and lowest DIC/TA ratio there. Strong biological $CO_2$ removal in the Mississippi River plume also contributes to low DIC/TA ratio and high $\Omega_{arag}$ values in local surface waters[42]. Finally, distinct from $pCO_2$ and pH, temperature-normalizing

$\Omega_{arag}$ does not alter the south-to-north spatial pattern in the Atlantic and GOM coasts (Fig. 3f).

**The California Current System.** The California Current System (CCS) extends from roughly the US-Canadian border to Baja California and is characterized by narrow shelves with strong, cold equatorward alongshore currents, and wind-driven upwelling events from late spring to early fall (Supplementary Fig. 1)[46–48]. While upwelling strength may vary, the observed carbonate chemical patterns are largely consistent (2007 and 2016 data were collected in late spring, while 2011, 2012, and 2013 data were collected in late summer) (Figs. 2 and 3). Here, similar to SSS, TA shows a weak south-to-north-decreasing gradient (Supplementary Table 3), while DIC has only a slightly stronger south-to-north-decreasing gradient[46]. As a result, there is no coherent meridional gradient in the DIC/TA ratio, reflecting different mechanisms at play from those along the Atlantic and Gulf coasts. However, higher $pCO_2$ and lower pH values are generally observed in the south.

As reported previously, $pCO_2$ is high and pH is low near coastal upwelling centers in the CCS (Fig. 3)[27,49]. These hot spots are caused by the strong upwelling of subsurface waters with high DIC/TA ratios and low $O_2$, pH, and $\Omega_{arag}$ values caused by biological respiration and anthropogenic $CO_2$ (Supplementary Fig. 5 and Table 3)[27,47]. An interesting characteristic of the CCS, offshore of the immediate upwelling centers, is that $pCO_2$ is generally lower and pH is higher relative to the Atlantic and GOM coasts at similar latitudes. This is mainly a result of the strong biological compensation following upwelling and incomplete compensation by air-sea gas exchange in the CCS as shown by field observations[49] and a numerical model[47]. However, once the temperature is normalized to 25 °C, such CCS vs. Atlantic contrasts largely disappear (Fig. 3). This reflects the important role of low SST values in keeping $pCO_2$ low and pH high via thermodynamic equilibrium shift and in maintaining a significant air–sea disequilibrium. In contrast, $\Omega_{arag}$ is almost ubiquitously low along the entire CCS coast relative to the GOM and southern Atlantic margins. It is particularly notable that, distinct from $pCO_2$ and pH, normalization to 25 °C does not alter this $\Omega_{arag}$ contrast between the West and East Coasts. Low SST in the CCS allows more $CO_2$ to be retained in the $CO_2$-rich upwelling waters due to a greater gas solubility, which keeps $[CO_3^{2-}]$ low (Eq. 1). Thus, the seemingly paradoxical co-existence of relatively high pH and low $\Omega_{arag}$ in the CCS reflects the distinct sensitivities of these parameters to thermodynamic constants, gas exchange, mixing, and biological processes.

**Other Pacific coastal regions.** The Gulf of Alaska has the widest continental shelves on the Pacific coast and is characterized by poleward winds and downwelling surface circulation during summer months[50]. Sea surface $pCO_2$ is low and pH is high in these relatively high-latitude, cold waters (Fig. 3), while $\Omega_{arag}$ is ubiquitously low. In Mexican Pacific coastal waters, south of the Gulf of California, SST is particularly high, while SSS and TA are similar to its northern neighbor (the CCS)[51]. Here, $pCO_2$ and pH values generally align with the weak south-to-north trends of Pacific coastal waters; however, DIC and DIC/TA are notably lower and $\Omega_{arag}$ is much higher than the waters to the north (Figs. 2 and 3).

**The Revelle factor distributions.** The capacity of seawater to resist a change in its acid–base properties in response to a DIC perturbation caused by $CO_2$ uptake from the atmosphere, or other processes, is important when considering the impacts of ocean acidification[9,10,52,53]. The characteristics of how each

carbonate system species will respond to such a disturbance can be described with a set of buffer factors[41,54,55]. Among them, the Revelle factor (RF) is most often used,

$$\text{RF} = \frac{\Delta pCO_2}{pCO_2} \Big/ \frac{\Delta \text{DIC}}{\text{DIC}}, \qquad (3)$$

which is the fractional change of the $CO_2$ species to a fractional DIC change at constant SST, SSS, and TA. A lower RF indicates a greater buffer capacity and a lower sensitivity of seawater $pCO_2$ to changes in DIC.

Spatial distributions of RF (Fig. 4a) show clear meridional gradients along the Atlantic margin, but no such pattern in the CCS. This suggests that the most buffered waters are in the GOM, southern US East Coast, and Mexican Pacific coast. The least buffered waters, which may be most sensitive to increasing ocean carbon content, are in the CCS upwelling centers and the mid to northern Atlantic waters of our study domain. In other words, the change in $pCO_2$, pH, and $\Omega_{arag}$ per unit DIC increase will be larger in the upwelling-dominated CCS waters and northern Atlantic waters than in other coastal waters of the North American margin. However, next to upwelled waters with characteristically high RF in the CCS, strong biological removal of $CO_2$ substantially increases the buffer capacity as indicated by the low RF values associated with high DO% and low nitrate at around 12 °C in Fig. 4c, d.

The spatial patterns of RF are similar and closely correlated to $[CO_3^{2-}]$ and $\Omega_{arag}$ (Figs. 3e, 4a, b and Supplementary Fig. 3 and Table 3), supporting the notion that the $CO_3^{2-}$ concentration plays a critical role in the ability to resist acidification as added anthropogenic or respiratory $CO_2$ is largely neutralized by $CO_3^{2-}$ to form $HCO_3^-$ (Eq. 2). Thus, $[CO_3^{2-}]$, $\Omega_{arag}$ and inversely the DIC/TA ratio (Fig. 2f) are good proxies or easy-to-understand surrogates for seawater buffer capacity. In all North American margin waters, the DIC/TA ratio is inversely correlated with $[CO_3^{2-}]$ and $\Omega_{arag}$ (Supplementary Table 3). In seawater, the point at which DIC/TA ≈ 1 (or DIC ≈ TA) represents the most sensitive state of the system to $CO_2$ addition because, at this point, $[CO_2] \approx [CO_3^{2-}]$ and $pH = -0.5 \log(K_1 K_2) - 0.5 \log([CO_2]/[CO_3^{2-}])$) changes the fastest[17]. Also note that this point (pH ≈ $-0.5 \log(K_1 K_2)$) occurs at a higher pH in cold waters (7.71 at 3 °C and 7.60 at 10 °C) than in warm waters (7.35 at 30 °C) (Supplementary Fig. 6), making cold waters more sensitive to $CO_2$ addition[56]. The cold waters of the northeastern margins, Alaskan shelf, and the CCS upwelling centers are approaching this point and are thus most vulnerable to ocean acidification.

**Continental-scale pattern vs. local variability.** To explore the first-order controlling mechanisms on large-scale spatial patterns of carbonate chemistry parameters on contrasting margins, we first compare field-observed DIC, $pCO_2$, pH, and $\Omega_{arag}$ distributions with those predicted from gas equilibrium with atmospheric $CO_2$. We do this by calculating each of these parameters from SST, SSS, and TA under the assumption of sea surface $CO_2$ gas equilibrium with the atmosphere. Over a large latitudinal range involving multiple water masses along the North American Atlantic and GOM coasts, DIC and $\Omega_{arag}$ generally follow air–sea gas equilibrium-based predictions on a seasonal timescale (months) (Fig. 5a, b, e, f). This atmospheric $CO_2$ equilibrium control mechanism would dictate a relatively high DIC and low $\Omega_{arag}$ in cold northern waters, where $CO_2$ solubility is high and the acid–base equilibrium favors the neutralization of $CO_2$ and $CO_3^{2-}$ to $HCO_3^-$ (Eqs. 1 and 2; Fig. 1a). In contrast, relatively low DIC and high $\Omega_{arag}$ conditions would occur in warmer southern waters, where $CO_2$ solubility and $K_1/K_2$ are lower, favoring $HCO_3^-$ dissociation to $CO_2$ and $CO_3^{2-}$ with $CO_2$ degassing to the

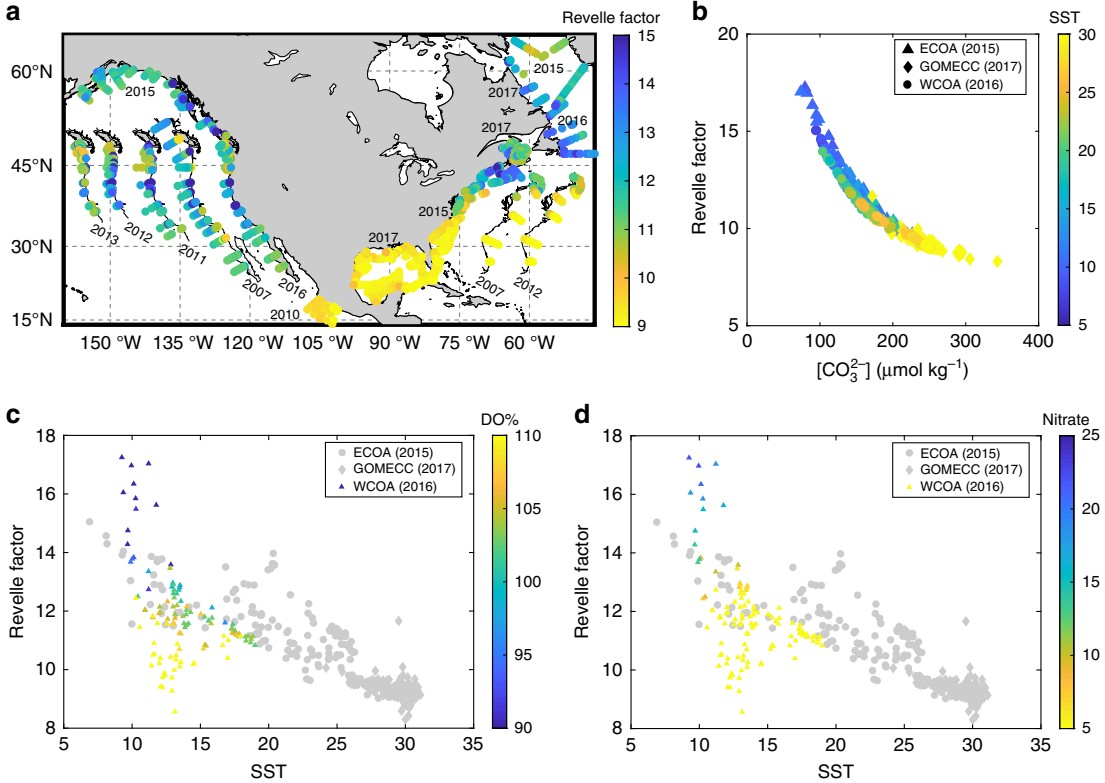

**Fig. 4 Revelle factor in the North Atlantic margins. a** Spatial distribution of the Revelle factor, **b** correlation of the Revelle factor to $[CO_3^{2-}]$, and **c**, **d** correlation of the Revelle factor to SST color coded by DO% and $[NO_3^-]$. Spatial distribution of $[CO_3^{2-}]$ is presented in Supplementary Fig. 4. In **c**, **d**, only the CCS data are color coded.

atmosphere. Such distribution patterns are known in global open-ocean basins[5–7,36,57] but have not been reported before in more dynamic coastal oceans. The fact that the DIC/TA ratio is lowest in the GOM and highest in northern Atlantic coastal waters reflects the temperature-regulated gas equilibrium and acid–base equilibrium shift. The observations of particularly low DIC/TA ratio and high $\Omega_{arag}$ in the Mexican Pacific margin (Figs. 2f and 3e) further support the argument made here that air–sea gas equilibrium largely controls DIC and $\Omega_{arag}$ in surface waters. This is true even in the region that is located right above the North Eastern Tropical Pacific (15–20°N, 100–110°W) oxygen mini-mum zone with elevated DIC concentrations of 2200 μmol kg$^{-1}$ at a depth of only 50 m (ref. [51]). To the north in the Gulf of Alaska, however, slightly lower DIC and higher $\Omega_{arag}$ than pre-dicted from atmospheric equilibration are likely due to legacy effects of strong biological $CO_2$ removal in late spring and early summer (Fig. 5c)[50].

Along the CCS, however, while atmospheric equilibrium still exerts strong control on DIC, upwelling and subsequent biological utilization of $CO_2$ have greatly modified the DIC concentrations in coastal waters. These processes have variable timescales of days to weeks[46,47,49] and are much less than the timescales of gas equilibration (one to a few months). The association of a positive DIC deviation with low temperature and high nutrients, a sign of upwelling (Fig. 5c, d), confirms that the source of DIC is from $CO_2$-rich subsurface waters that have been upwelled in the CCS. The impact of upwelling and biology becomes greater for $\Omega_{arag}$, which deviates substantially from the atmospheric equilibrium-based prediction in the CCS (Fig. 5g, h).

Despite the strong role of air–sea gas exchange, observed pH and $pCO_2$ values can deviate greatly from those at atmospheric equilibrium even in the Atlantic and GOM coasts (Fig. 5i, j, m, n). This is particularly true in the CCS where strong upwelling (high

nutrient) or biological production (low nutrient) lead to extreme pH and $pCO_2$ values, whereas the expected pH and $pCO_2$ from the atmospheric equilibrium are nearly invariable (Fig. 5k, l, o, p). Deviations of DIC, $\Omega_{arag}$, and pH from gas equilibration are highly correlated with positive or negative deviations of seawater $pCO_2$ from atmospheric $pCO_2$ on all margins (Supplementary Fig. 7). In the CCS, the positive deviation in DIC and negative deviations in $\Omega_{arag}$ and pH are associated with $CO_2$ super-saturation induced by the upwelling of $CO_2$-rich subsurface waters, while the opposite conditions ($CO_2$ undersaturation) are associated with the subsequent stimulation of biological production[27,47]. In the GOM, such deviations are controlled by biological $CO_2$ removal in the low-salinity river plume and net respiration and $CaCO_3$ precipitation in the high-salinity waters. On the Atlantic coast, a combination of temperature changes associated with physical transport by large-scale alongshore currents, biological $CO_2$ removal, and local terrestrial exports are all at play (Supplementary Fig. 7).

**Sensitivity of carbonate chemistry to perturbations.** A funda-mental question to ask is why do DIC and $\Omega_{arag}$ generally follow air–sea gas equilibrium-based predictions in the Atlantic and Gulf coasts, while pH and $pCO_2$ distributions are decoupled from the atmospheric equilibrium and are most sensitive to local physical and biological modifications? We contend that this is mainly the result of two factors. First and most importantly, the temperature effect on carbonate chemistry has two components, one is the internal effect on the thermodynamic equilibrium constants, including the acid–base dissociation constants $K_1$ and $K_2$ and the solubility constant $K_0$ under no gas exchange conditions, while the other is the external effect related to DIC concentration changes caused by gas exchange. As shown in Fig. 1, in an open

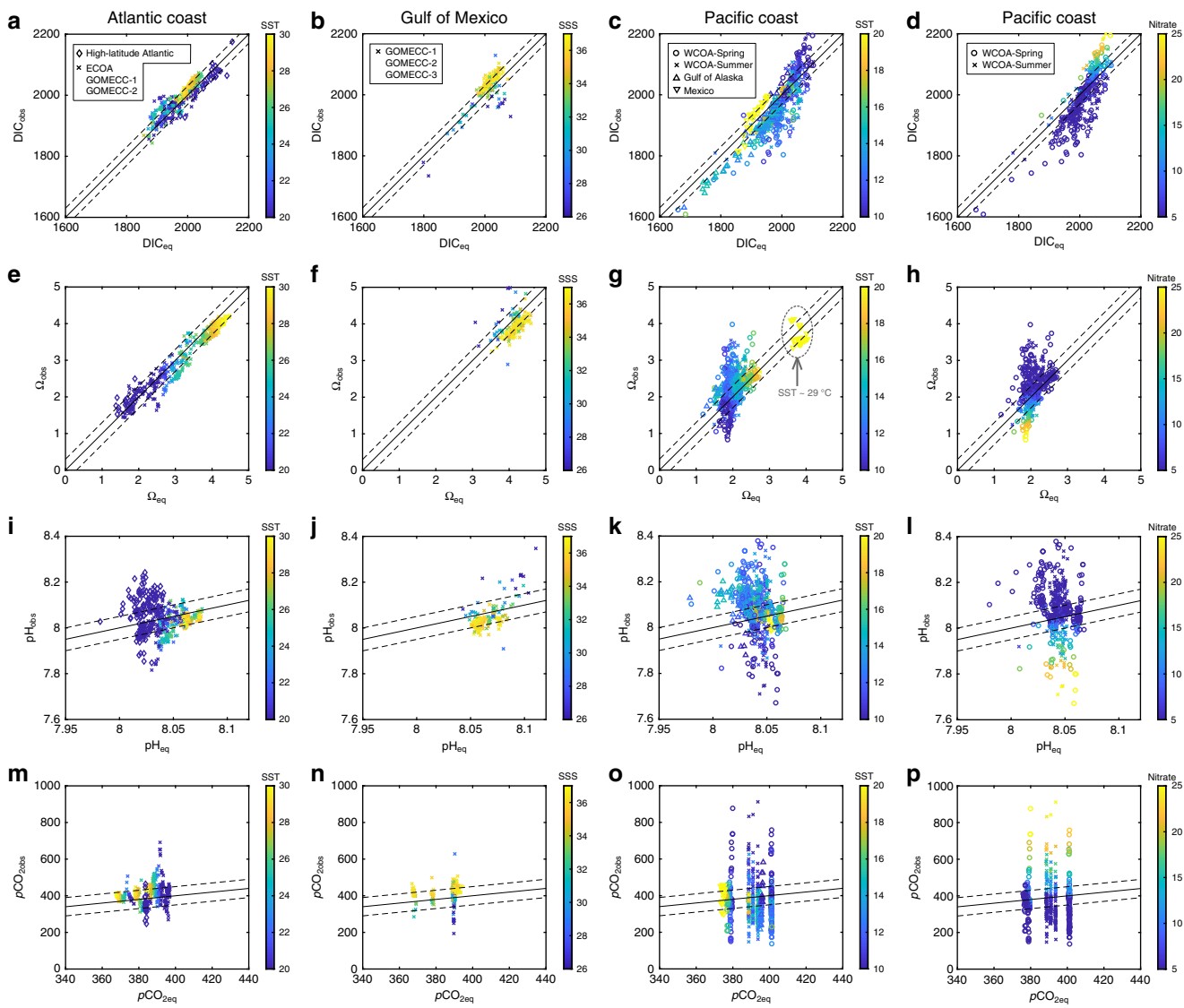

**Fig. 5 Observations vs. calculated values.** These values are at equilibrium with atmospheric $CO_2$ along North American margins. **a–d** are DIC, **e–h** are $\Omega_{arag}$, **i–l** are pH, **m–p** are $pCO_2$ in the Atlantic Coast (column 1), Gulf of Mexico (column 2), and Pacific Coast (columns 3 and 4). The Pacific Coast occupies two columns color coded by SST and Nitrate, respectively. $DIC_{obs}$, $\Omega_{obs}$, $pH_{obs}$, and $pCO_{2obs}$ stand for the measured DIC and properties calculated from measured DIC and TA. $DIC_{eq}$, $\Omega_{eq}$, $pH_{eq}$, and $pCO_{2eq}$ represent values calculated under the conditions of seawater equilibrium with atmospheric $pCO_2$ at the time of field observation programs (see the "Methods" section). $pH_{obs}$ and $pH_{eq}$ are on the total proton concentration scale. The dashed lines around the 1:1 line are $\pm 30 \ \mu mol \ kg^{-1}$ for DIC, $\pm 0.3$ for $\Omega$, $\pm 0.05$ for pH, and $\pm 50 \ \mu atm$ for $pCO_2$.

system these two temperature effects tend to cancel each other out completely for pH and $pCO_2$ and partially for $[CO_2]$, but have an additive effect for $[CO_3^{2-}]$, DIC, and $\Omega_{arag}$, with the gas equilibrium playing the dominant role. Thus, moderate carbonate system perturbations caused by local physical and biological processes are essentially overshadowed for DIC and $\Omega_{arag}$ due to their large latitudinal gradients along the Atlantic and GOM margins associated with the south-to-north temperature gradient, but are apparent for pH and $pCO_2$, which have no clear meridional trends. The contrasting temperature effects are also compounded by the fact that physical and biological addition and removal of DIC act linearly, while their impacts on $pCO_2$ changes are non-linear. Second, while the internal temperature effect and upwelling of high DIC subsurface waters are essentially instantaneous or of short timescale, air–sea gas exchange compensates carbonate chemistry changes over monthly timescales. As a result, slow-acting surface processes such as cooling in poleward-moving currents along the Atlantic coast are largely compensated and

appear to have relatively small impacts on surface seawater carbonate chemistry. The processes that contribute most visibly to carbonate chemistry, in particular $pCO_2$ and pH variability, are those that act much faster than the timescales of gas equilibrium (e.g., upwelling, river input, and primary production, which lack SST-related meridional trends)[36].

These factors also contribute to the observed sensitivity differences of pH and $\Omega_{arag}$ to anthropogenic, physical, and biological processes in the CCS upwelling waters. In these waters, excess $CO_2$ relative to $CO_2$ expected from equilibration with a preindustrial atmosphere comes from both subsurface organic carbon respiration and anthropogenic $CO_2$ obtained when subsurface water masses were previously in contact with the atmosphere. These sources can combine to create seawater that greatly exceeds even the anthropogenically elevated modern atmospheric $pCO_2$ and cause pH and $\Omega_{arag}$ in the newly upwelled waters to be much lower than they would be at atmospheric equilibrium (Supplementary Fig. 7). Subsequent $CO_2$ degassing

and biological blooms can reduce $CO_2$ and increase pH and $\Omega_{arag}$. A very strong correlation between DO and $\Omega_{arag}$ in the CCS (Supplementary Table 3a) suggests that biological drawdown of DIC and upwelling of waters rich in respired DIC together overpower the influence of the small latitudinal SST-gradient in this region. However, generally lower temperatures in the CCS (compared to the GOM and south Atlantic regions) and incomplete compensation by gas exchange allow these waters to simultaneously have relatively high pH but low $\Omega_{arag}$ because the effects of low temperature outweigh the impacts of biological $CO_2$ removal for $\Omega_{arag}$ but not for pH (Supplementary Fig. 8). The same discussion applies to high-latitude waters in the Labrador Sea in the Atlantic and Gulf of Alaska in the Pacific all have high pH but low $\Omega_{arag}$, although in these places, the main driver for a low $\Omega_{arag}$ is the high $CO_2$ solubility at low SSTs and the main driver for a high pH is the biological $CO_2$ removal.

**Timescales of coastal ocean processes.** To support the conclusions derived from the field data and first principle-based analysis and to further explore the different responses of carbonate parameters to atmospheric forcing and local physical and biological processes, we simulated seasonal changes in carbonate chemistry using a time-variable box model for the Atlantic coast. The model simulates carbonate parameter changes and the associated gas exchange flux in an idealized surface mixed layer following prescribed time series of salinity, temperature, wind, biological production, and vertical exchange with the subsurface water (see details in the "Methods" section). In the northeastern margin, SST follows a typical seasonal cycle, while SSS is high in winter and decreases to a minimum in summer as a result of changing river discharge and meltwater supply[37] (Fig. 6a). As expected, surface water $pCO_2$ increases and pH decreases from spring to summer following the seasonal warming and display opposite behavior from summer to fall following cooling (Fig. 6b). This thermal seasonality is enhanced by high riverine $CO_2$ export and reduced by spring biological blooms[35]. The $pCO_2$ decrease and pH increase in the summer to fall cooling period are also enhanced by biological blooms and decreased river discharge. In the meantime, in contrast, DIC increases toward winter and decreases toward summer, whereas $\Omega_{arag}$ decreases toward winter and increases toward summer (Fig. 6c).

For comparison with the box model simulation, we also calculate carbonate system parameters while assuming instantaneous equilibrium between sea surface $pCO_2$ and the atmosphere (open system conditions). The full-model Atlantic margin $pCO_2$ and pH values completely decouple from the values expected from atmospheric equilibrium, but track water mass property changes associated with seasonal temperature variations and short-term local physical and biological processes. Note that the equilibrium values exhibit a much smaller range of $pCO_2$ and pH variability due to cancellation of the two temperature effects under the imposed conditions where gas exchange is instantaneous (Fig. 1 and Fig. 6b, d, e). In contrast, the modeled DIC and $\Omega_{arag}$ values approximately track those predicted by instantaneous equilibrium with atmospheric $CO_2$, although with a time lag of 1–2 months. Here a seasonal DIC maximum and $\Omega_{arag}$ minimum occur during winter when $CO_2$ solubility is the highest with opposing results found during summer (Fig. 6c, f, g). These model-derived behaviors are consistent with the field observations and our mechanistic interpretation of them; although seasonal SST variations are used to illustrate the carbonate chemistry behaviors in the simple box model, latitudinal SST variations are reflected in the summertime observations along the Atlantic and Gulf coasts.

Interestingly, modeled $pCO_2$ and pH values also show the high-frequency signals of SST and SSS variations caused by local short-term processes reflected in SST and SSS changes. In contrast, the impacts of high-frequency local processes are largely absent in the modeled DIC and greatly dampened in modeled $\Omega_{arag}$ values (Fig. 6). When a disturbance (vertical mixing with high DIC subsurface water) is introduced (with a sudden increase of DIC), with all other conditions unchanged, the $e$-folding time of $CO_2$ system restoration (that is the disturbance is reduced to $1/e$ or 37% of the initial value) due to degassing varies from 10 days at very high gas exchange rates (to mimic an instantaneous gas equilibration) to more than 100 days at low gas exchange rates (approaching a closed system) with ~1 month at an average wind condition and typical mixed-layer depths (Supplementary Fig. 9). In general, two to three $e$-folding intervals are needed for a disturbed signal to be completely erased. Increases in the associated nutrient input, and thus net biological production, will have an opposite effect and speed up the restoration (not shown). Thus the very different carbonate chemistry behaviors reported in this work reflect timescales of physical, chemical, and biological processes differ from instantaneous thermodynamic equilibrium, including rapid local physical processes (upwelling, mixing, etc.) acting on timescales of days to weeks, biological production (weeks), alongshore current transport (months), gas exchange processes (months to seasonal), and the seasonal thermal cycle.

## Discussion

Our study illustrates that, as in the open ocean, the large-scale meridional gradient in surface water temperature regulates $CO_2$ gas equilibration with the atmosphere and plays a first-order role in determining the spatial distributions of DIC and $\Omega_{arag}$ on seasonal timescales in non-upwelling-dominated ocean margins. In contrast, $pCO_2$ and pH variations are more reflective of short-term, local modifications by coastal ocean physical and biological processes. In upwelling-dominated regions along eastern boundary current ocean margins, however, the atmospheric $CO_2$ equilibrium mechanism is perturbed much more than that along other ocean margins and in the open ocean[5,6,8,36,58]. This is because dynamic coastal conditions further accentuate the contrasts between longer timescale air–sea equilibrium in slowly moving coastal currents and modifications by short-term, local physical and biological processes. As such, the two commonly used ocean acidification metrics, pH and $\Omega_{arag}$, can vary quite differently in upwelling vs. non-upwelling-dominated systems and in warm southern vs. cold northern waters. As organisms respond to pH and $\Omega_{arag}$ differently (both during development and calcification)[14], our findings emphasize the importance of examining multiple aspects of organismal and ecosystem responses to ocean acidification with respect to $pCO_2$, pH, $\Omega_{arag}$, and DIC changes under warmer and higher $CO_2$ future ocean conditions.

Recently, contrasting seasonal cycle responses of $\Omega_{arag}$, $[H^+]$, pH, and $pCO_2$ to DIC increases throughout the global surface ocean have been shown with observations[10] and models[9,11,13,59]. These findings, in addition to a more localized, estuarine study[60], suggest that seasonal (and higher-frequency) variability may intensify more rapidly on upwelling-dominated and cold high-latitude margins due to their naturally lower buffering capacity and higher sensitivity to increasing $CO_2$ content in the ocean. However, the absolute rates of $\Omega_{arag}$ decrease are faster in warm and high $\Omega_{arag}$ tropical and subtropical waters[26]. Our work further shows that there are substantial spatial differences in buffering capacities across the North American ocean margins and these differences imply a natural disparity in regional sensitivities

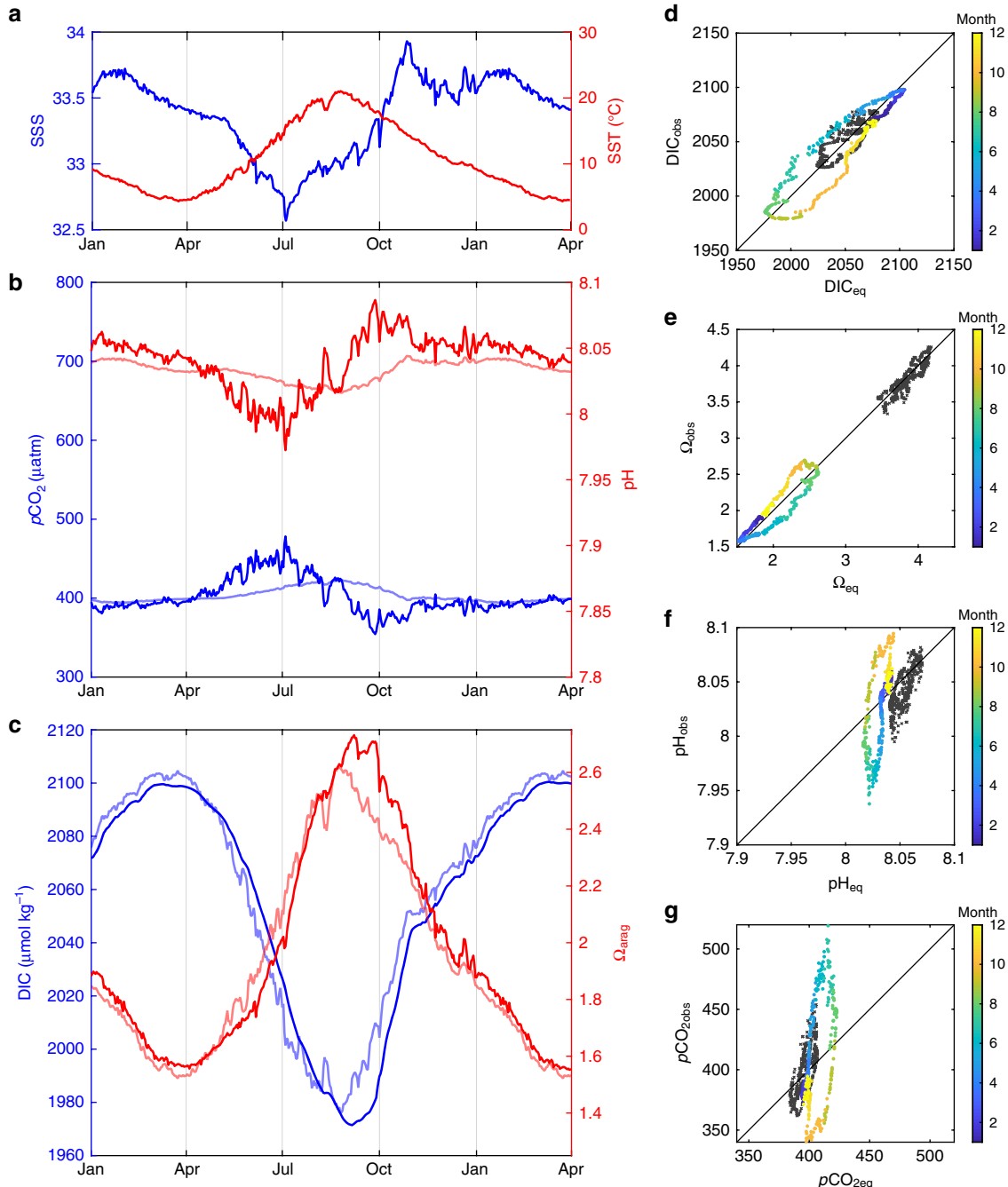

**Fig. 6 Simulations of *p*CO₂, pH, DIC, and Ω_arag.** Simulations were run using a box model and values derived from the assumption of instantaneous CO₂ equilibrium with the atmosphere. In the left column are **a** prescribed SST and SSS, **b** model-generated *p*CO₂ and pH, and **c** model-generated DIC and Ω_arag for Gulf of Maine waters. Heavy lines are box-model simulations and light lines are calculated values from instantaneous equilibrium with the atmosphere. For simplicity, we assume atmospheric *p*CO₂ is seasonally invariant. In the right column, model observed **d** *p*CO₂, **e** pH, **f** DIC, and **g** Ω_arag are plotted against those predicted from equilibration with the atmosphere (eq), similar to those in Fig. 5. Symbols color coded by month represent the Gulf of Maine simulation (with colder temperature), and black symbols represent the South Atlantic Bight simulation (warmer temperature).

to ocean acidification. In particular, the RF distribution clearly draws our attention to ecosystems—including the CCS and northern-latitude coastal regions, such as the Gulf of Maine (Atlantic) and the Gulf of Alaska (Pacific)—as being particularly sensitive and vulnerable to anthropogenic CO₂ forcing.

## Methods
**Description of the field program.** Starting in 2007, under the auspices of National Oceanic and Atmospheric Administration (NOAA)'s Climate Program Office and, since its inception in 2010, Ocean Acidification Program, NOAA Pacific Marine

Environmental Laboratory and NOAA Atlantic Oceanographic and Meteorological Laboratory, in collaboration with academic partners and international collaborators, have conducted multiple surveys of the North American ocean margins. The most recent quality-controlled survey data are from the *E*ast Coast Ocean Acidification (ECOA) and Gulf of Alaska cruises in summer 2015, the West Coast Ocean Acidification (WCOA) cruise in late spring 2016, and the GOM Ecosystem and Carbon Cycle (GOMECC) cruise in summer 2017. Results from these cruises are presented in Fig. 2 and 3 along the coastlines of the North American continent and associated ocean margins, while previous cruises in the same regions (West Coast: late spring 2007 and late summer 2011, 2012, and 2013; GOM and Atlantic: summer 2007 and 2012). In addition, data from the Scotian Shelf and Labrador Sea as well as the Mexico Pacific coast, although limited, are included to extend the

geographic coverage. Supplementary Table 2 summarizes the relevant cruise information.

In this work, all DIC samples were measured at sea by coulometric titration using a modified Single-Operator Multi-Metabolic Analyzer system, and TA was measured by acidimetric titration using the open cell method[61,62]. Both DIC and TA measurements were calibrated daily using Certified Reference Materials from Dr. Andrew Dickson's laboratory at Scripps Institution of Oceanography. The overall uncertainty of each of these two measurements is $\pm 2$ µmol $kg^{-1}$. DO was measured by a Winkler titration technique. Detailed descriptions are available at, for ECOA 2015, [https://www.nodc.noaa.gov/oads/data/0159428.xml] or [https://doi.org/10.7289/v5vt1q40], for WCOA 2016, [https://www.nodc.noaa.gov/oads/data/0169412.xml] or [https://doi.org/10.7289/v5v40shg], for GOMECC 2017, [https://www.nodc.noaa.gov/oads/data/0188978.xml] or [https://doi.org/10.25921/yy5k-dw60], and for AZMP, [https://www.dfo-mpo.gc.ca/science/data-donnees/biochem/index-eng.html].

For the purpose of examining spatial distributions, pH, $pCO_2$, and $\Omega_{arag}$ were calculated from DIC and TA together with nutrient concentrations using the CO2SYS program[63] applying the carbonic acid dissociation constants of Mehrbach et al.[64] as refitted by Dickson and Millero[65]. We use the DIC–TA pair because both parameters are available and were measured at high quality during all cruises. Good internal consistency among multiple parameters was observed and potential issues were identified in an earlier study[66]. pH is expressed on the total proton concentration scale[67]. $\Omega_{arag}$ is defined as the concentration product of dissolved calcium and carbonate ions divided by the aragonite mineral solubility product[68].

**DIC, $\Omega_{arag}$, pH, and $pCO_2$ at equilibrium with atmospheric $pCO_2$.** The values at equilibrium are calculated assuming that seawater $pCO_2$ is in equilibrium with annual mean atmospheric $pCO_2$. In this case, the input pair to CO2SYS is $pCO_2$ and TA. The atmospheric $CO_2$ data, as a dry air mole fraction, was obtained from the Mauna Loa Observatory, Hawaii.

**Box-model simulation.** To study the relationships that control surface-ocean carbonate parameters and air–sea $CO_2$ fluxes, we use a box model to simulate an idealized mixed layer. The box model simulates the carbonate systems as a homogeneous surface-ocean water mass located either in the North-West Atlantic (NWA, 40.1–42.5°N, 69.2–67.5°W) and South Atlantic Bight (SAB, 28.3°N–29.7°N, 77.6°W–79.2°W). The model is driven by realistic temperatures and salinities from the Mercator 1/12° data-assimilated General Circulation Model with a daily resolution in time[69].

The carbonate system is defined from DIC and TA and re-adjusted to changes in temperature and salinity at each time step using the CO2SYS package[63]. Throughout the model simulation, TA is calculated from prescribed salinity[61,69]. The model is initiated with the carbonate system defined by $pCO_2$ in equilibrium with atmospheric values and TA ($TA_{t0}$) defined by initial salinity[70]. DIC is calculated from $pCO_{2(aq)}$, $TA_{t0}$, temperature, and salinity at $t = 0$. The model is spun up for 180 days using prescribed physical and biological conditions as described above. $pCO_{2\_t(aq)}$ is calculated using CO2SYS as well. $\Delta DIC_{Bio}$ and $\Delta DIC_{Vertical}$ are prescribed (see below), while $\Delta pCO_{2\_air–sea}$ is calculated at each time step using the relationship $\Delta pCO_{2\_air–sea} = 0.24 \times k \times K_0 \times (pCO_{2\_t(aq)} - (pCO_{2\_(atm)}))$. $K_0$ is the $CO_2$ gas solubility[71] and $k$ is defined as $k = 0.251 \times W^2 \times (Sc/660)^{-0.5}$, where $W$ is wind speed in m $s^{-1}$ and $Sc$ is the Schmidt number[72,73].

Biological production is assessed from weekly averages of daily changes in satellite-derived chlorophyll (Chl) for the year 2015 using the daily MODIS aqua 4-km level 3 product[74]. We extract daily time series for each grid cell that falls within the NWA and SAB model regions, identify all pairs of consecutive days with valid data, and convert changes in Chl to a carbon flux by using a fixed C:Chl ratio of 60 (ref. [75]). The resulting weekly averages are interpolated to daily time series. The resulting change in DIC due to biological production is significantly smaller than other sources and sinks in the current study. The highest net community production (NCP) values we get are on the order of 0.1 mmol C $day^{-1}$ $m^{-3}$. This corresponds to ~50 mg C $m^{-2}$ $day^{-1}$. Satellite-derived net primary production (NPP) is on the order of 400 mg C $m^{-2}$ $day^{-1}$ and with an expected NCP to NPP ratio of ~10%; thus, our biological production parameters are reasonable.

Winds are prescribed to 7 m $s^{-1}$ in the summer and 10.5 m $s^{-1}$ in the winter, to be consistent with representative wind data from the NOAA/Seawinds blended wind data set[69]. Winter mixing in NWA is simulated by adding 0.1 mmol $m^{-3}$ DIC daily from October to February (over 155 days). The value is based on a scaling analysis of vertical DIC gradients and diffusivity estimates in the region based on observational in 2015 in the NWA region. Literature values of physical vertical diffusion combined with vertical profiles of DIC from the ECOA 2015 cruise suggests that diffusive transports of DIC is negligible during summer. We use constant winds to minimize noise and to isolate the effect by changes in solubility on the carbonate system. Atmospheric $pCO_2$ is set to 395 µatm.

Specifically, the forward iteration from time $t\_n$ to $t\_n + 1$ is performed by the following steps:

$$DIC_{t+1} = DIC_t + \Delta pCO_{2\_air–sea} + \Delta DIC_{Bio} + \Delta DIC_{Vertical}.$$

1. Calculate the carbonate system defined by temperature, salinity, DIC, and TA at $t\_n$.

2. Apply changes to the DIC concentration at $t\_n + 1$ by biological processes, vertical mixing and air–sea exchange (based on $pCO_2$ values at $t\_n$).
3. Recalculate the carbonate system using temperature, salinity, and TA at $t\_n$, and DIC at $t\_n + 1$.
4. Record pH, $pCO_2$, and $\Omega_{arag}$ at $t\_n + 1$.

The reason to use temperature, salinity, and TA at $t\_n$ when performing step 3 is to make sure that all carbon parameters are calculated using the same physical conditions. All the details are given inside the coding: https://doi.org/10.5281/zenodo.3833678.

We also calculate $DIC_{eq}$, a property that is based on the same TA, salinity, and temperature as the model but with $pCO_{2(aq)}$ relaxed to $pCO_{2(atm)}$, which can be interpreted as the air–sea exchange being infinitely fast. Here for simplicity, we assume $pCO_{2(atm)}$ equals the dry $CO_2$ mole fraction.

**Statistical information.** We used the built-in Matlab function (corrcoef) to compute the correlation coefficients (https://www.mathworks.com/help/matlab/ref/corrcoef.htl).

## Data availability

Discrete bottle data from all NOAA ocean acidification regional research cruises are available at NOAA's National Centers for Environmental Information. For ECOA 2015, [https://www.nodc.noaa.gov/oads/data/0159428.xml] or [https://doi.org/10.7289/v5vt1q40]; for WCOA 2016, [https://www.nodc.noaa.gov/oads/data/0169412.xml] or [https://doi.org/10.7289/v5v40shg]; for GOMECC 2017, [https://www.nodc.noaa.gov/oads/data/0188978.xml] or [https://doi.org/10.25921/yy5k-dw60], and for AZMP, [https://www.dfo-mpo.gc.ca/science/data-donnees/biochem/index-eng.html]. Atmospheric $CO_2$ data are available at https://www.esrl.noaa.gov/gmd/ccgg/trends/data.html.

## Code availability

The box model was written in python and the code can be found at https://doi.org/10.5281/zenodo.3833678.

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

## Acknowledgements

Data reported in this work were collected under the auspices of the National Oceanic and Atmospheric Administration (NOAA)'s Global Carbon Cycle (GCC) and Ocean Acidification Programs (OAP). We thank captains and crew of all vessels involved in collecting the data at sea and the outstanding technical staff, students, postdocs, and PIs at NOAA and partner labs for their collaboration on the first decade of NOAA ocean acidification cruises. The Canadian data were collected with support from NSF and Department of Fisheries and Oceans (DFO) Aquatic Climate Change Adaptation Services Program (ACCASP). We are grateful to the governments of Canada and the United Mexican States for fishing licenses and research permission allowing us to conduct our research within their Exclusive Economic Zones during all cruises (PPFE/DGOPA-072/16, PPFE/DGOPA-137/17, EG0082017, EG00032016, No. 343609, and IDR-1279). W.-J.C was also supported by NSF (grant no OCE-1559279). B.J. was supported by Simons Foundation via grant no 549947 (SS). This is contribution number 4853 from the NOAA Pacific Marine Environmental Laboratory.

## Author contributions

R.A.F. and S.R.A. are responsible for the design of the West Coast Ocean Acidification (WCOA) cruises. J.C. is responsible for the design of the Gulf of Alaska Coast Ocean Acidification cruises. J.E.S., W.-J.C., and R.W. are responsible for the design of the East Coast Ocean Acidification (ECOA) cruises. R.W. and L.B. are responsible for the design of the Gulf of Mexico Ecosystem and Carbon Cycle (GOMECC) cruises. K.A.-S. and P.P. are responsible for the Davis Strait work. J.M.H.-A. is response for the Mexican Pacific coastal data. L.-Q.J. compiled the initial list of cruise data. J.J.R., B.C., N.H., Q.L. and C.L. contributed to data collection efforts. Y.-Y.X. and W.-J.C. analyzed data. W.-J.C. prepared the paper. B.J. did the box-model simulation. L.X. and W.-J.C. did the temperature effect model. A.J.F., S.R.A., B.R.C., R.W., R.A.F., B.J., A.J.S., C.-T.A.C., and D.K.G. edited the paper. All authors contributed to the discussion and writing.

## Competing interests

The authors declare no competing interests.
