## [Peer Review File · Nature Communications]

Reviewers' comments:

Reviewer #1 (Remarks to the Author):

This review is for manuscript NCOMMS-19-24366, entitled "Controls on Surface Water Carbonate Chemistry along North American Ocean Margins", by Wei-Jun Cai et al. The paper compiles data from multiple coastal cruises conducted along all of the North American coastlines, and goes on to diagnose the processes which set distributions of pH, pCO₂, and aragonite saturation state. While the data quality and quantity are truly impressive, I feel that the discussion and interpretation of these data are better suited for a more specialized journal. The reasons for my decision are threefold:

- 1) Carbonate chemistry relationships are not straightforward for a general audience, and their presentation is complicated by the many-paneled figures both in the main text and in the Supplement.
- 2) The manuscript is long, and at times contains what seem to be trivial and/or self-evident conclusions about covarying properties of the carbonate system.
- 3) The main conclusions of the manuscript are, in the end, qualitative, and therefore the relevance falls short of the quality necessary for publication in Nature Communications. Given the nature of the audience, may be much better suited to a specialized oceanographic journal.

To address the first point, many of the panels of the main text figures, certainly, can be removed to get the main points across. In fact, it may only be appropriate to display the panels which show significant correlations (i.e. high r-squared values in Table S3). In Figure 1, H⁺ and pH are redundant, as are [CO₃] and aragonite saturation. The definition of "gas equilibrium" in Figure 1 I also believe is incorrect. The authors call this term the difference between the open and closed system cases, but in fact, if you do the math I believe it is the difference between three terms, in other words:

$$GE = OS(T) - CS(T) + OS(30),$$

where GE is the gas equilibrium term, OS(30) is the reference value of the derived parameter at 30C, and OS,CS(T) are the open,closed system calculated values at a given temperature T respectively. This is the only way that I can get, for instance, a pCO₂ "difference" of almost 700 ppm between the open system (395) and closed system(~100) cases at 3C. The "difference" of 700 ppm makes no sense -- these calculations must be made more explicit, and explained more clearly. I also recommend the authors use lines instead of points for this figure.

Figure 2 is very busy. The authors should think carefully about which terms are crucial for the interpretation of their data, and only present those that are most meaningful to the discussion. The rest should be moved to the Supplemental. I am also not sure of the utility of Figure 3, given the calculations done between closed- and open system cases in Figures 4 and 5. I am also unconvinced that Figure 5 needs to be in the main text.

Figure 7 feels very redundant, because all buffer factors appear to have a similar relationship with carbonate ion concentration. The entire figure could be reduced to one panel, or removed entirely, with Figure 8 showing the relevant information to the Buffer capacity section. While the point made by the authors is valid, I feel that many of the relationships shown in Figure 7 are self-evident as they are mainly defined by conservative carbonate system parameters (TA and DIC, plus acid dissociation constants), and the equations governing the carbonate system (see the authors' definitions), and are therefore trivial.

This last paragraph brings me to the second point -- many relationships, for instance the correlation between TA/DIC and saturation state, appear trivial, because they fall readily out of the definitions of the carbonate system. For instance, the correlation of Revelle Factor with carbonate ion concentration is because of the definition of the Revelle Factor, and the fact that carbonate ion is two protons away from (i.e. quadratically related to) pCO₂. Secondly, because the Revelle Factor is primarily defined by conservative properties, it should not necessarily scale with temperature, as shown by the overlapping warm and cold points in Figure 7. It is therefore not surprising at all that DIC/TA is a good proxy for seawater buffering capacity anywhere in the ocean, or in any body of water for that matter. The

authors should make sure that any conclusions drawn from their data are not simply pre-determined relationships between calculated parameters in the carbonate system. This issue may have come about by trying to explain these concepts to a more general audience -- another reason why this manuscript may be better suited in a more technical journal.

Finally, while the box model appears useful for showing the qualitative conclusions drawn from the data, it could be used to be much more quantitative. It does a very nice job, in my opinion, of showing the difference between the equilibration of DIC on seasonal timescales, and the lack of equilibration of pCO₂. But, what ARE the timescales at play here? The box model could be used to test timescales of biological uptake, air-sea equilibration, and upwelling, to show how fast these processes need to be acting for the system to be in or out of balance. Quantifying these timescales at play in different coastal systems might elevate this manuscript to being worthy of publication in this journal. However, the model is poorly described in the Methods -- only one equation with little explanation -- and poorly cited. They refer to time series of biological production and vertical flux (of what?), yet do not cite these time series. Biological production is somehow related to daily changes in POC in two boxes, yet the only boxes that are explicitly defined are a mixed layer and an atmosphere (which cannot, I think, have any POC!). Mixed layer thickness is not defined and is somehow parameterized as vertical flux. A schematic should be provided of the box model; even more appropriate would be providing the box model code. Because this is a time-evolving box model, the authors have the ability to diagnose exactly the timescales responsible for setting, for instance, the equilibration of DIC versus pCO₂.

Minor comments:

Abstract

Lines 48-50: This conclusion does not seem strong enough. Which coastal ocean processes are at play? What are the major drivers of deviations from equilibrium? What are their timescales?

Main text

Line 68: more, not longer, or -- longer timeseries

Line 76: What about the influence of low riverine salinity on [Ca] and therefore on the solubility of solid carbonates?

Line 83-84: Last sentence is awkward.

Lines 88-90: Equation punctuation is incorrect. Should be CO₂ uptake: ...and a period at the end of the equation. New sentence starts with Where...

Line 91: first and second dissociations constants of carbonic acid, respectively.

Lines 97-109: See comment above about "gas equilibrium." This term must be defined more clearly and I am not convinced its magnitude is always correct, specifically in cases where the sense of change is opposite for closed and open-system cases.

Lines 108-109: Is it your model? These appear to be the basic equations governing carbonate chemistry, in cases of open- and closed- systems.

Line 110: Why quotations around "open system", "closed system", and "gas equilibrium"? These appear to be self-explanatory terms and therefore do not need quotations.

Line 113: This is misleading, as pCO₂ is held constant by definition in the open system case.

Line 126: These changes are not global but...continental-scale? margin-scale?

Lines 173-174: This point seems to again be a function of the relationship between DIC/TA (i.e. conservative properties) and the mathematical calculation of [CO₃]

Lines 180-181: Isn't this true only at the very mouth of the river, as stated in the previous paragraph?

Lines 195-196: Awkward sentence; rephrase.

Line 201: Why the use of "in situ"?

Lines 204-205: What do you mean by "or are modified"?

Lines 217-224: This paragraph feels more descriptive and may be more appropriate earlier in this Pacific Coast section.

Line 225: Again, this is not necessarily global control, but continent-scale or margin-scale...

Lines 225-235: Figure 4 is very effective; more so than Figure 3. I would consider removing Figure 3,

or moving it to the supplement.

Lines 235-236: HCO_3^- dissociates on a timescale of acid-base equilibration, i.e. on the timescale of aqueous proton diffusion. Therefore the use of "readily dissociates" is misleading. Consider rephrasing this sentence.

Lines 244-245: what are the timescales of air-sea CO_2 exchange and biological production?

Lines 245-247: Where is the NETP OMZ? Give lat and long, so we can reference it on the maps.

Lines 254-257: Cannot clearly see the Gulf of Alaska points on this plot. They are hidden under the other data.

Lines 321-352: See above comment about buffer capacities. Much of this section, along with Figure 7, feels redundant and/or trivial.

Line 360-362: Again, feels trivial/unnecessary.

Line 363: I think that you switched western and eastern.

Line 772: Who did the box modeling?

Reviewer #2 (Remarks to the Author):

Understanding drivers of coastal carbonate chemistry is important for better predicting changes due to anthropogenic CO_2 . This paper uses extensive cruise data from multiple years to describe drivers (mostly physical) of coastal carbonate chemistry. I believe that this is an important contribution to the oceanography field.

My comments for improvement are relatively minor.

Line 65: Coastal carbonate chemistry work has also focused quite a bit on the effects of community composition and community metabolism as drivers of carbonate dynamics and potential to buffer or exacerbate OA.

e.g., Silbiger and Sorte 2018 Sci Reports, Lowe et al. 2019 Sci Reports, Kleypas et al. 2011 Global Change Biology, Muehllehner et al. 2016 Global Biogeochemical cycles

Remove the several leading phrases such as "in what follows", "as will be discussed", "see further explanation later", etc.

Also, when citing a figure there is no need to also direct to the figure caption (e.g., See Fig 1, and its caption).

Paragraph 72: Submarine groundwater discharge is also an important process that can affect coastal seawater carbonate chemistry (and can often exceed riverine input).

e.g., Nelson et al. 2015 Marine Chemistry, Cyronak et al. 2013 Biogeosciences, Cyronak et al. 2014 Global Biogeochem. Cycles; Wang et al. 2014 Environ. Sci. Technol.

Line 127: define "coastal" (e.g., within the 200m isobaths?)

Line 138: Some kind of correlation analysis across the 3 years would make this statement stronger.

Line 172: add citation after (metric for OA stress on organisms)

Line 629: cruise to cruises

Methods: What program did you use to statistically analyze your data (i.e correlation coefficients)

Data availability: I highly encourage the authors to make their code readily available along with the

data for transparent and reproducible science.

The lines in figure 4 are really hard to see

Reviewer #3 (Remarks to the Author):

REVIEW OF "Controls on Surface Water Carbonate Chemistry along North American Ocean Margins" by Cai et al. submitted for publication in Nature Communications.

This manuscript deals with an relevant issue about the carbon cycle in the coastal ocean: the sensitivity of the different components of the CO₂ system to main controlling factors (also related to general stressors of the ocean): warming and anthropogenic input. It presents a synthesis of the surface data from a wonderful full column data base of hydrographic and chemical data in the North American Ocean margins.

Being potentially a manuscript suitable for Nature Communications it suffers mainly from a lack of focus and blurred methodology which both make the ms difficult to follow. I really think that it can be much improved and submitted again. My general recommendation is MAJOR REVISION.

One of the main and longer (also difficult to follow) parts of the ms deals with the sensitivity of the CO₂ system to temperature changes in case of a close system (not in contact to the atmosphere) and an open system (equilibrated with a nominal pCO₂ in the atmosphere). Those sensitivities are explained in Fig 1, Sup Fig 2, Sup Fig 5, Sup Table 1. I tried to reproduce Fig 1 myself using CO₂SYN and I made it with the following matlab script

```
DIC=1956.5;
TA=2300;

T=(5:1:30)';
Sal=35;

[Aclose,head]=CO2SYS(TA,DIC,1,2,Sal,T,T,0,0,1,0.1,1,4,1);
Aopen=CO2SYS(TA,395,1,4,Sal,T,T,0,0,1,0.1,1,4,1);
Ind=[18, 23, 19,22,31,21,2]';

Titles=head(Ind,:);

figure
for ii=1:size(Ind,1)
    subplot(4,2,ii+1)
    plot(T,Aclose(:,Ind(ii)),'bo')
    hold on
    plot(T,Aopen(:,Ind(ii)),'g*')
    plot(T,(Aclose(26,Ind(ii))+Aopen(:,Ind(ii))-
Aclose(:,Ind(ii))), 'r*')
    grid on
    title(Titles(ii,:))

end
legend('closed','open','gas equil')
```

The gas equilibrium effect is obtained starting from an open system equal to a close system at 30°C. However if the DIC/TA ratio is changed, note that $1956.5/2300 = 0.85$, a very low value, so a system quite buffered. If TA = 2100, DIC/TA= 0.93 and the results for Figure 1 would be very different. I do not think Fig 1 should be within the main text, in fact, it is very conf using for me. Instead I do agree to keep Sup Fig 2 and Sup Fig 5 to show the thermal sensitivity of the different CO₂ system variables within a closed and open system.

The first paragraphs before the Results section are quite cumbersome, discouraging the reader to keep reading the work. I suggest to shorten them with a brief explanation linked to Sup Fig 2 and Sup Fig 5 about the thermal sensitivities. Then a introduction to the data set linked to Sup Fig 1 and Sup Table 2 and the main findings of the results.

RESULTS section

This section is very descriptive and it could benefit from a small reorganization of the figures that could probably smooth the reading:

Fig 2 h-pCO₂ in situ, i-pH is, j- OmegaAragonite is could be moved to Figure 3.

Fig 3 would have on the left the in situ values for pCO₂, pH and Omega and on the right those normalized to 25°C.

Fig 2 in order to have a pair number of figures could include surface NO₃ as plot h.

Please state in all figures .. that you are showing surface data, and clearly refer if in situ or normalized to 25°C.

As far as I understand the data showed in Fig 5 is also contained in Fig 4, as the CCS is part of the Pacific Coast. I suggest adding another column to Fig 4 so that one shows the Pacific coast without the CCS, then the CCS, the GOM and the Atlantic coast.

I would suggest moving Fig 7 .. also showing surface distributions of the buffers factors after Figure 3.. and the corresponding paragraph too.

In the section Global influence versus local variability is mainly focused on the CCS, leaving very little explanations to GOM or the Atlantic coast.

The Sensitivity of carbonate chemistry to perturbations section would benefit from moving the Buffer capacity distribution section upwards. In the sensitivity section the authors already talk about the buffer capacity theory.

DISCUSSION

This part of the ms is mostly an abstract, it does not read as a discussion. I suggest to speculate on main processes to be altered by Global Change affecting the North America Coast regions: warming, higher or lower upwelling, lower river discharging in the GOM or higher river discharging plus ice melting in the North, changes in the near coast currents and so on.

I hope to have been helpful.

Reviewer #1 (Remarks to the Author):

This review is for manuscript NCOMMS-19-24366, entitled "Controls on Surface Water Carbonate Chemistry along North American Ocean Margins", by Wei-Jun Cai et al. The paper compiles data from multiple coastal cruises conducted along all of the North American coastlines, and goes on to diagnose the processes which set distributions of pH, pCO₂, and aragonite saturation state. While the data quality and quantity are truly impressive, I feel that the discussion and interpretation of these data are better suited for a more specialized journal. The reasons for my decision are threefold:

- 1) Carbonate chemistry relationships are not straightforward for a general audience, and their presentation is complicated by the many-paneled figures both in the main text and in the Supplement.
- 2) The manuscript is long, and at times contains what seem to be trivial and/or self-evident conclusions about covarying properties of the carbonate system.
- 3) The main conclusions of the manuscript are, in the end, qualitative, and therefore the relevance falls short of the quality necessary for publication in Nature Communications. Given the nature of the audience, may be much better suited to a specialized oceanographic journal.

Response: First we thank the reviewer's appreciation of the quality and quantity of the data presented here. We also agree with the reviewer that carbonate chemistry relationships are not straightforward for a general audience. However, the current intense interest of our community, other relevant communities, and the public regarding ocean acidification and its biological and ecological consequences has elevated the subject to be suitable for high-impact journals (and indeed there are many recent publications in high-impact journals on this and related topics). Our paper is the first of its kind presenting a chemical dataset that encompasses all coastal, marine margins of an entire continent (except the Arctic Ocean margins) and explaining the key drivers. Our style of presenting all major features and explaining these features from first principles contributes to the more theoretical nature of the discussion. It was our intention to thoroughly address what drives carbonate chemistry relationships along different ocean margins by taking advantage of our large dataset and the long format of *Nature Communications* articles (upper to 5000 words & 9 figures), as well as by targeting a broader audience (beyond a few carbonate chemists). This is also why the foundation of the marine carbonate chemistry is reviewed and some of the seemingly trivial points (e.g., the several buffer factors and their subtle differences) are included. In the revision, we try to strike a balance between our intention and the reviewer's suggestions to focus on the few most important issues (really just two: temperature responses and timescales). In particular, we reduced the buffer factor discussion to include only the Revelle Factor. Our revision aims to balance the input from this and the other two reviewers.

To address the first point, many of the panels of the main text figures, certainly, can be removed to get the main points across. In fact, it may only be appropriate to display the panels which show significant correlations (i.e. high r-squared values in Table S3). In Figure 1, H⁺ and pH are redundant, as are [CO₃] and aragonite saturation. The definition of "gas equilibrium" in Figure 1 I also believe is incorrect. The authors call this term the difference between the open and closed system cases, but in fact, if you do the math I believe it is the difference between three terms, in other words:

$$GE = OS(T) - CS(T) + OS(30),$$

where GE is the gas equilibrium term, OS(30) is the reference value of the derived parameter at 30C, and OS,CS(T) are the open,closed system calculated values at a given temperature T respectively. This is

the only way that I can get, for instance, a pCO₂ "difference" of almost 700 ppm between the open system (395) and closed system (~100) cases at 3C. The "difference" of 700 ppm makes no sense -- these calculations must be made more explicit, and explained more clearly. I also recommend the authors use lines instead of points for this figure.

Response: The reviewer is correct in spelling out exactly how we did the calculation on the "gas exchange" term: $GE = OS(T) - CS(T) + OS(30)$. However, the open system value at 30C is a fixed reference point for both OS and CS conditions, so does not affect their relative patterns. This reference point can be at any temperature. Choosing 30C is for the illustration and convenience but it makes the closed system results appear "unrealistic" though thermodynamically they are correct. In the revision, we modify this by starting with a moderate temperature (e.g., choose a reference point of 18C) and imposing both warming and cooling, though thermodynamically both approaches are correct. We also moved the thermodynamic constants into Fig. 1 to make it easier for readers to comprehend. All points are replaced by lines in Fig. 1.

In the revision, we also take the reviewer's suggestion to remove several slightly redundant panels (such as between [H⁺] and pH, between [CO₃²⁻] and omega, etc.). We also take the suggestion by Reviewer #3 to move pCO₂, pH and Omega to Fig. 3 (as left column) for comparison with temperature normalized pCO₂, pH and Omega (as right column). Thus, Fig. 2 now has fewer panels and Fig. 3 provides a nice contrast between variables at the field temperature (SST) and 25C.

While in principle we would agree with the reviewer that "it may only be appropriate to display the panels which show significant correlations (i.e. high r-squared values in Table S3)," in this case, we respectfully disagree. We feel in the case of a pair of closely related parameters X and Z (e.g., pH and omega), when X correlates with parameter Y, but Z doesn't correlate with Y because of a good reason, then it makes sense to present both spatial distributions of X(Y) and Z(Y) to explain this point. Also see below.

Figure 2 is very busy. The authors should think carefully about which terms are crucial for the interpretation of their data, and only present those that are most meaningful to the discussion. The rest should be moved to the Supplemental. I am also not sure of the utility of Figure 3, given the calculations done between closed- and open system cases in Figures 4 and 5. I am also unconvinced that Figure 5 needs to be in the main text.

Response: We agree with the reviewer that Fig. 2 is very busy. However, we present all these individual panels to make a point that different parameters respond to regional temperature differently. For example, Omega has a strong correlation with SST while pH has no correlation with SST and there is a good reason why this is so. If we only present Omega and leave out pH, this point cannot be conveyed effectively. Another example is "there is no coherent meridional gradient in the DIC/TA ratio in the CCS, reflecting the existence of a different control mechanism there than that along the Atlantic coast." In this case, it won't make sense to present DIC/TA ratio only on the Atlantic coast but not on the Pacific coast only because the latter has no significant correlation with SSS or SST or other parameter. To make Fig. 2 less busy and more readable as recommended by reviewer #1, and also to implement reviewer #3's good suggestion, we now re-organize Fig. 2 by moving 2i, 2j and 2k to the left column of Fig. 3 to contrast with the temperature normalized plots in the right column. Now Fig. 2 is less busy.

By moving in situ pCO₂, pH and Omega from Fig. 2 to Fig. 3, the contrasting property distributions at in situ and 25C are clearer. Thus, we have kept Fig.3. We merged Fig.5 and Fig. 4 (it is now new Fig. 5) to show that in the CCS, upwelling and biological production (with high and low nitrate) regulate the deviation from the predicted values.

Figure 7 feels very redundant, because all buffer factors appear to have a similar relationship with carbonate ion concentration. The entire figure could be reduced to one panel, or removed entirely, with Figure 8 showing the relevant information to the Buffer capacity section. While the point made by the authors is valid, I feel that many of the relationships shown in Figure 7 are self-evident as they are mainly defined by conservative carbonate system parameters (TA and DIC, plus acid dissociation constants), and the equations governing the carbonate system (see the authors' definitions), and are therefore trivial.

Response: We agree that Fig. 7 feels somewhat redundant. While the set of buffer factors have some subtle differences and these differences can help to explain the different sensitivities of various species to the disturbance of increasing atmospheric CO₂, we agree that presenting these subtle differences probably distracted focus from the main story line of this paper (temperature responses of various species and timescales of various processes). Thus, we have reduced the buffer capacity presentation to only the Revelle Factor and have taken the reviewer's suggestion to simplify Fig. 7 and combine it with Fig. 8. Reviewer #3 suggests we move this figure earlier to the Results (as it is a result), and we agreed. So, it is now the new Fig. 4.

This last paragraph brings me to the second point -- many relationships, for instance the correlation between TA/DIC and saturation state, appear trivial, because they fall readily out of the definitions of the carbonate system. For instance, the correlation of Revelle Factor with carbonate ion concentration is because of the definition of the Revelle Factor, and the fact that carbonate ion is two protons away from (i.e. quadratically related to) pCO₂. Secondly, because the Revelle Factor is primarily defined by conservative properties, it should not necessarily scale with temperature, as shown by the overlapping warm and cold points in Figure 7. It is therefore not surprising at all that DIC/TA is a good proxy for seawater buffering capacity anywhere in the ocean, or in any body of water for that matter. The authors should make sure that any conclusions drawn from their data are not simply pre-determined relationships between calculated parameters in the carbonate system.

This issue may have come about by trying to explain these concepts to a more general audience -- another reason why this manuscript may be better suited in a more technical journal.

Response: We certainly agree with the reviewer and are well aware that marine carbonate system has only two freedoms, that is from any two parameters, one can derive all other properties provided that the total concentrations of other acid-base (i.e., PO₄³⁻ and organic acids) and all acid dissociation constants are known. However, we feel it doesn't help to simply declare "that is the way it is." Instead, we feel it will serve the ocean acidification community well to explain some of the first principle-based predictions (e.g., that pH and Omega respond very differently to meridional temperature gradient). That said, we heed the reviewer's feedback and have modified our tone in presenting some of the results. In particular, we greatly reduced old Fig. 7 and combined it with old Fig. 8 (now as the new Fig. 4).

Finally, while the box model appears useful for showing the qualitative conclusions drawn from the data, it could be used to be much more quantitative. It does a very nice job, in my opinion, of showing the

difference between the equilibration of DIC on seasonal timescales, and the lack of equilibration of pCO₂. But, what ARE the timescales at play here? The box model could be used to test timescales of biological uptake, air-sea equilibration, and upwelling, to show how fast these processes need to be acting for the system to be in or out of balance. Quantifying these timescales at play in different coastal systems might elevate this manuscript to being worthy of publication in this journal. However, the model is poorly described in the Methods -- only one equation with little explanation -- and poorly cited. They refer to time series of biological production and vertical flux (of what?), yet do not cite these time series. Biological production is somehow related to daily changes in POC in two boxes, yes the only boxes that are explicitly defined are a mixed layer and an atmosphere (which cannot, I think, have any POC!). Mixed layer thickness is not defined and is somehow parameterized as vertical flux. A schematic should be provided of the box model; even more appropriate would be providing the box model code. Because this is a time-evolving box model, the authors have the ability to diagnose exactly the timescales responsible for setting, for instance, the equilibration of DIC versus pCO₂.

Response: Thank you for your appreciation of the time variable box model. The original purpose here was to use the box model to support our explanations of the observation. Now we take the reviewer's suggestion to expand the box model description and emphasize on illustrating the time scales at play. We do this only for the Atlantic coast. For the CCS, we took results from the very nice physical and biological coupled model simulations presented in Turi et al. 2014 and field observations (e.g., Hales et al. 2005). The discussion of timescales at play supports and strengthens our conclusions as well as our discussion of first principles in earlier sections. We thank the reviewer for the suggestion.

Minor comments:

Abstract

Lines 48-50: This conclusion does not seem strong enough. Which coastal ocean processes are at play? What are the major drivers of deviations from equilibrium? What are their timescales?

<the abstract is consolidated, in particular this sentence in the last part is removed.>

Main text

Line 68: more, not longer, or -- longer timeseries

<yes, change to more time; the sentence is removed in the final revision>

Line 76: What about the influence of low riverine salinity on [Ca] and therefore on the solubility of solid carbonates?

<added a brief mention of this, line 80, "In the low salinity zone, rivers can also reduce carbonate saturation state by reducing [Ca²⁺].">

Line 83-84: Last sentence is awkward.

<we agree with you and delete the sentence. We later come back to this point in the Results.>

Lines 88-90: Equation punctuation is incorrect. Should be CO₂ uptake: ...and a period at the end of the equation. New sentence starts with Where...

<correction made as suggested.>

Line 91: first and second dissociations constants of carbonic acid, respectively.

<correction made as suggested. >

Lines 97-109: See comment above about "gas equilibrium." This term must be defined more clearly and I am not convinced its magnitude is always correct, specifically in cases where the sense of change is opposite for closed and open-system cases.

<see our earlier response. As suggested by the reviewer, this term is now clearly defined in the figure caption. The large difference between the closed and open systems at very low temperature is simply because the selection of a very high temperature, 30C, as the reference point, which leads to a very large temperature difference and thus a very low pCO₂ in the closed system. While there is nothing wrong in such illustrative calculation and selection of reference point is arbitrary, we change the reference point to 18C to avoid that impression of excessively large pCO₂ difference between the open and closed systems. >

Lines 108-109: Is it your model? These appear to be the basic equations governing carbonate chemistry, in cases of open- and closed- systems.

<No, it is not our model. How about change to framework? But one may argue it is not our framework either. So, we change it to a more neutral or less grandiose word "in our theoretical consideration.">

Line 110: Why quotations around "open system", "closed system", and "gas equilibrium"? These appear to be self-explanatory terms and therefore do not need quotations.

<We now remove the quotations around open system and closed system. We also change to term "gas equilibrium" to "gas exchange," which is more general and includes both exchange and the final equilibrium. Here "gas exchange" should not be interpreted as a kinetic term. It is only used here as is defined in Fig. 1 caption. Occasionally, we also use the term "equilibrium with the atmosphere" to avoid confusion with gas exchange kinetics.>

Line 113: This is misleading, as pCO₂ is held constant by definition in the open system case.

<Yes, it is by definition. But this is also true in the real world as atmospheric pCO₂ change is negligible comparing to those in waters. So we add a modifier "by definition no change in pCO₂">.

Line 126: These changes are not global but...continental-scale? margin-scale?

<Here in the term "global and regional ocean warming and acidification", by "global" we meant a general trend applicable to global scale (that is overall global ocean temperature is increasing and pH is decreasing). However, locally/regionally these trends can be different. To be more general in the Introduction, we removed both modifiers and simplified this term to only "ocean warming and acidification.">

Lines 173-174: This point seems to again be a function of the relationship between DIC/TA (i.e. conservative properties) and the mathematical calculation of [CO₃]

<Yes, this is true. This is why in the next sentence we say “ Ω_{arag} and its associated parameters,...” While we totally agree that in the CO₂ system, we can calculate all parameters from two observed, we find it useful to relay why these parameters show such different distributions, e.g., pH and pCO₂ distributions are totally different than omega and DIC.>

Lines 180-181: Isn't this true only at the very mouth of the river, as stated in the previous paragraph?

<Actually, the strongest biological CO₂ removal occurs at intermediate salinity ($S = 25-33$) as shown in Guo et al. 2013 (ref# 42). But in our 2017 GOM survey, the large research vessel didn't reach the very nearshore truly low salinity areas. Thus, here near the lowest salinity $S=30$ is considered as “low salinity” plume waters.>

Lines 195-196: Awkward sentence; rephrase.

<sentence is modified>

Line 201: Why the use of "in situ"?

<We agree that “in situ” is not an appropriate term and is now deleted. It was intended to differentiate observed (at “in situ” SST) and temperature-normalized distributions. We now use “at field temperature” or just “SST” whenever it is needed to differentiate a property with its temperature normalized distribution.>

Lines 204-205: What do you mean by "or are modified"?

<Now deleted. We intended to say the pattern disappeared or is altered. >

Lines 217-224: This paragraph feels more descriptive and may be more appropriate earlier in this Pacific Coast section.

<We now separate California Current System, CCS, and the rest of the Pacific coasts into two sections; so, this part stands alone and won't feel like new results after a discussion of the CCS observation.>

Line 225: Again, this is not necessarily global control, but continent-scale or margin-scale...

Lines 225-235: Figure 4 is very effective; more so than Figure 3. I would consider removing Figure 3, or moving it to the supplement.

<Thank you for the appreciation of Fig. 4. We have incorporated Fig. 5 into Fig. 4 to become the new Fig. 5. We also moved field pH, pCO₂ and omega from Fig. 2 into Fig. 3 to be together with the temperature normalized pH, pCO₂ and omega as suggested by another reviewer. We think this is a good approach. Also when we say “global” we often only mean domain-wide as in contrast to various locations within the domain. But we agree in this case, we should use continental-scale instead.>

Lines 235-236: HCO₃⁻ by dissociates on a timescale of acid-base equilibration, i.e. on the timescale of aqueous proton diffusion. Therefore the use of "readily dissociates" is misleading. Consider rephrasing this sentence.

< By “readily” we didn’t mean timescale or kinetics, but rather an equilibrium tendency. But we agree and modified it to “equilibrium moves to the left in Eq. 1”. >

Lines 244-245: what are the timescales of air-sea CO₂ exchange and biological production?

<Air-sea CO₂ exchange timescale varies between one and a few months depending largely on the mixed layer depth while biological production timescales are generally on the order of days to weeks. Timescales of physical transportation and vertical mixing also are different on different margins. In the revised paper, we add here and there the general timescales of various processes.>

Lines 245-247: Where is the NETP OMZ? Give lat and long, so we can reference it on the maps.

<added>

Lines 254-257: Cannot clearly see the Gulf of Alaska points on this plot. They are hidden under the other data.

<Yes, many are hidden under the other data, but all the triangle symbols are below the 1:1 line>

Lines 321-352: See above comment about buffer capacities. Much of this section, along with Figure 7, feels redundant and/or trivial.

<See earlier response. We now greatly simplified this part and moved it to Results.>

Line 360-362: Again, feels trivial/unnecessary.

<We understand the reviewer’s feeling about this issue and probably can also admit that, on the fundamental theory side, we offer nothing really new to the acid-base equilibrium and buffer theory. However, one of the purposes of this paper is to use the large dataset in contrasting margins to explain carbonate chemical or acid-base equilibrium properties under partial gas equilibrium with the atmosphere to a large audience. We try to serve that purpose here.>

Line 363: I think that you switched western and eastern.

<We are correct in our description of “eastern boundary current ocean margins (e.g., CCS)” and “western boundary current margins (e.g., Gulf Stream).” (Here they refer to eastern Pacific Ocean and western Atlantic Ocean boundary current, respectively.) In the revision, we deleted “western boundary current margins.” Instead, we feel using the name “non-upwelling dominated ocean margins” is sufficient.>

Line 772: Who did the box modeling?

<Thanks for reminding us. We now added “B. Jönsson did the box-model simulation” in the Author Contribution.>

Reviewer #2 (Remarks to the Author):

Understanding drivers of coastal carbonate chemistry is important for better predicting changes due to

anthropogenic CO₂. This paper uses extensive cruise data from multiple years to describe drivers (mostly physical) of coastal carbonate chemistry. I believe that this is an important contribution to the oceanography field.

<We appreciate this positive comment>

My comments for improvement are relatively minor.

Line 65: Coastal carbonate chemistry work has also focused quite a bit on the effects of community composition and community metabolism as drivers of carbonate dynamics and potential to buffer or exacerbate OA.

e.g., Silbiger and Sorte 2018 Sci Reports, Lowe et al. 2019 Sci Reports, Kleypas et al. 2011 Global Change Biology, Muehllehner et al. 2016 Global Biogeochemical cycles

<Agreed and added this in the introduction with the most recent ref Silbiger and Sorte 2016 and the earlier Kleypas et al. 2011. Thanks!>

Remove the several leading phrases such as “in what follows”, “as will be discussed”, “see further explanation later”, etc.

<done>

Also, when citing a figure there is no need to also direct to the figure caption (e.g., See Fig 1, and its caption).

<agreed and done>

Paragraph 72: Submarine groundwater discharge is also an important process that can affect coastal seawater carbonate chemistry (and can often exceed riverine input).

e.g., Nelson et al. 2015 Marine Chemistry, Cyronak et al. 2013 Biogeosciences, Cyronak et al. 2014 Global Biogeochem. Cycles; Wang et al. 2014 Environ. Sci. Technol.

<Though we believe these are very local examples, we are happy to add a phrase into the text. Also by virtue of the design of the NOAA coastal OA cruises, we'd be very unlikely to see these effects anyway, as the ships would not get close enough to shore to detect these inputs. >

Line 127: define “coastal” (e.g., within the 200m isobaths?)

<In this paper we include all data collected by OAP program cruises and other similar cruise roughly to 1000m depth and 400km from shoreline (this sentence is now added to the Methods section). We intend to include the impacted zone beyond 200m as our “coastal zone”, thus it is wider than the 200m continental shelf. This was suggested in the NACP synthesis workshop (Hales, Cai, Sabine... 2008) and was adopted by SOCAT for coastal ocean CO₂ data archive. >

Line 138: Some kind of correlation analysis across the 3 years would make this statement stronger.

<We agree. We added such an analysis of the 2015 against 2012 data and 2015 against 2007 in the Atlantic coast in a new Table 3. Both are highly correlated. As we cannot do the spatial cross-correlation using the original data due to different cruise tracks and station locations. The scattered data were interpolated to cover the same region before running the cross-correlation.>

Line 172: add citation after (metric for OA stress on organisms)

<Agreed. There are many we can cite. But since we will exceed the maximum of 70, we will only add a few already cited in this paper.>

Line 629: cruse to cruise

<We have corrected this. Thanks.>

Methods: What program did you use to statistically analyze your data (i.e correlation coefficients)

<We used the built-in Matlab function "corrcoef" to compute the correlation coefficients (<https://www.mathworks.com/help/matlab/ref/corrcoef.html>). We now add this information to the Methods.>

Data availability: I highly encourage the authors to make their code readily available along with the data for transparent and reproducible science.

<Thanks for the suggestion and we agree. Now added>

The lines in figure 4 are really hard to see

<This is probably due to the low-resolution figures in the review version. We now try to reduce the white space as much as possible and to enlarge the labels to improve the readability.>

Reviewer #3 (Remarks to the Author):

REVIEW OF "Controls on Surface Water Carbonate Chemistry along North American Ocean Margins" by Cai et al. submitted for publication in Nature Communications.

This manuscript deals with an relevant issue about the carbon cycle in the coastal ocean: the sensitivity of the different components of the CO₂ system to main controlling factors (also related to general stressors of the ocean): warming and anthropogenic input. It presents a synthesis of the surface data from a wonderful full column data base of hydrographic and chemical data in the North American Ocean margins.

Being potentially a manuscript suitable for Nature Communications it suffers mainly from a lack of focus and blurred methodology which both make the ms difficult to follow. I really think that it can be much improved and submitted again. My general recommendation is MAJOR REVISION.

<We appreciate very much you constructive and adoptable suggestions. We now focus on two main issues: the thermal sensitivities and the timescales of processes. We have simplified the buffer factor section significantly. We also largely followed your suggestion to reorganize the figures, while also using fewer figures overall, as suggested by another reviewer>.

One of the main and longer (also difficult to follow) parts of the ms deals with the sensitivity of the CO₂ system to temperature changes in case of a close system (not in contact to the atmosphere) and an open system (equilibrated with a nominal pCO₂ in the atmosphere). Those sensitivities are explained in Fig 1, Sup Fig 2, Sup Fig 5, Sup Table 1. I tried to reproduce Fig 1 myself using CO₂SYN and I made it with the following matlab script

```
DIC=1956.5;
TA=2300;
T=(5:1:30)';
Sal=35;
[Aclose,head]=CO2SYS(TA,DIC,1,2,Sal,T,T,0,0,1,0.1,1,4,1);
Aopen=CO2SYS(TA,395,1,4,Sal,T,T,0,0,1,0.1,1,4,1);
Ind=[18, 23, 19,22,31,21,2]';
Titles=head(Ind,:);
figure
```

```

for ii=1:size(Ind,1)
subplot(4,2,ii+1)
plot(T,Aclose(:,Ind(ii)), 'bo')
hold on
plot(T,Aopen(:,Ind(ii)), 'g*')
plot(T, (Aclose(26,Ind(ii))+(Aopen(:,Ind(ii))-
Aclose(:,Ind(ii))))), 'r*')
grid on
title(Titles(ii,:))
end
legend('closed', 'open', 'gas equil')

```

The gas equilibrium effect is obtained starting from an open system equal to a close system at 30°C. However if the DIC/TA ratio is changed, note that $1956.5/2300 = 0.85$, a very low value, so a system quite buffered. If TA = 2100, DIC/TA= 0.93 and the results for Figure 1 would be very different. I do not think Fig 1 should be within the main text, in fact, it is very conf using for me. Instead I do agree to keep Sup Fig 2 and Sup Fig 5 to show the thermal sensitivity of the different CO2 system variables within a closed and open system.

The first paragraphs before the Results section are quite cumbersome, discouraging the reader to keep reading the work. I suggest to shorter them with a brief explanation linked to Sup Fig 2 and Sup Fig 5 about the thermal sensitivities. Then a introduction to the data set linked to Sup Fig 1 and Sup Table 2 and the main findings of the results.

<Thanks for the comment and suggestion. You are correct in how we did the illustrative calculation. We agree that essentially Sup Fig. 2 explains all the thermal sensitivities. However, since one of the two focuses of this paper is to use the thermal sensitivities to explain the contrasting property distributions, we feel it is best to discuss this in the Introduction section. Also, the intention of this paper is to use a new dataset from contrasting coastal ocean margins around the North American margins to explain carbonate chemistry principles to a larger audience. Therefore, we feel a somewhat unusual manuscript style, with a longer introduction, is appropriate to set the stage for the follow up Results and Discussion. However, based on your feedback, we have worked to streamline the presentation and make it easier to follow. We also bring Sup Fig. 2 into a simpler Fig. 1 (but moved a few panels in the old Fig. 2 to Supplement).>

RESULTS section

This section is very descriptive and it could benefit from a small reorganization of the figures that could probably smooth the reading:

<Thank you for these good suggestions. We largely followed this restructuring of figures.>

Fig 2 h-pCO2 in situ, i-pH is, j- OmegaAragonite is could be moved to Figure 3.

<yes, done>

Fig 3 would have on the left the in situ values for pCO2, pH and Omega and on the right those normalized to 25°C.

<yes, done>

Fig 2 in order to have a pair number of figures could include surface NO3 as plot h.

Please state in all figures .. that you are showing surface data, and clearly refer if in situ or normalized to 25°C.

<We decided to move some of the panels into a supplementary figure. So Fig. 2 is now much simpler.>

As far as I understand the data showed in Fig 5 is also contained in Fig 4, as the CCS is part of the Pacific Coast. I suggest adding another column to Fig 4 so that one shows the Pacific coast without

the CCS, then the CCS, the GOM and the Atlantic coast.

<Thanks for the suggestion, we agree. We combined Fig. 4 and Fig. 5 as the new Fig. 5>.

I would suggest moving Fig 7 .. also showing surface distributions of the buffers factors after Figure 3.. and the corresponding paragraph too.

<We agree and combined a simplified version of Fig. 7 and the old Fig. 8 as the new Fig. 4. However, we present only Revelle Factor to discuss buffer capacity.>

In the section Global influence versus local variability is mainly focused on the CCS, leaving very little explanations to GOM or the Atlantic coast.

<This section is intended to explain the large-scale (or domain-wide) pattern vs. local variability. The large-scale pattern is the fact the DIC and Omega along the Atlantic coast agree with those calculated from the atmospheric equilibrium. Both Atlantic and Pacific coasts have local variability but the CCS has the strongest variability. We modified language to make this part clearer.>

The Sensitivity of carbonate chemistry to perturbations section would benefit from moving the Buffer capacity distribution section upwards. In the sensitivity section the authors already talk about the buffer capacity theory.

<We agree and have moved the buffer capacity section earlier in the Results. We now only present and discuss the Revelle Factor. This simplification will make the presentation easier to follow and allows us to focus more on the main story line.>

DISCUSSION

This part of the ms is mostly an abstract, it does not read as a discussion. I suggest to speculate on main processes to be altered by Global Change affecting the North America Coast regions: warming, higher or lower upwelling, lower river discharging in the GOM or higher river discharging plus ice melting in the North, changes in the near coast currents and so on.

<Your sense about the Discussion section “not read as a discussion” is correct. But this is the house style of *Nature* journals: you present results and you discuss their meaning right away to avoid duplication. Thus, most discussion is included in the Results section. The Discussion section (normally limited to two paragraphs) is limited to the discussion of general implications (we did this at the end of the first paragraph and in the entire second paragraph). But your suggestions are good. Thus, we simplified and shortened the more “summary/abstract” sentences in the first paragraph and tried to add words and sentences toward implications. Thanks.>

I hope to have been helpful.

<You have been very helpful! As a result we feel the paper is easier to follow and the quality is improved. Again thank you very much for all the very helpful suggestions.>

REVIEWERS' COMMENTS:

Reviewer #1 (Remarks to the Author):

This review is for the manuscript entitled "Controls on Surface Water Carbonate Chemistry along North American Ocean Margins" by Wei-Jun Cai et al., which I reviewed in its previous form. I am pleased to say that the manuscript has been much improved, and I recommend it for publication with minor revisions. The main text is more streamlined, and the discussion of timescales is very nice and indeed shores up the authors' conclusions, from seasonal timescales on upwards. I include the comments below, most of which I believe the authors could take or leave. However, at the end, I think it may be worth mentioning the Revelle Factor analysis more strongly in the conclusions. This may help to drive the point home that northern-latitude coastal ecosystems, both in the Pacific and Atlantic sectors, are particularly vulnerable to anthropogenic CO₂ forcings.

Detailed comments:

TEXT

Abstract:

Lines 49-51: this is good.

Introduction:

Line 72: I would put a sentence here that states this section's aims, i.e. constructing a theoretical framework to understand temperature effects on the carbonate system. Right now I feel like the point is a little buried.

Line 73-85: This paragraph feels a bit lost. I'm not sure it belongs here as it disrupts the temperature discussion. Perhaps it belongs down in the final Discussion, or placed somewhere else in the text?

Results:

Lines 159-161: I think you need to make the point of doing the 25C calculation much more clear. What do you hope to see by comparing the SST and 25C-normalized plots? What is the usefulness of doing such a comparison? Spelling this out will make this whole section flow better, I think.

Lines 181-183: Have you thought about the role of calcification in modifying Omega (and carbonate ion) along the coasts? I am surprised this doesn't feature at all in the discussion, actually, given the threat to calcifiers of OA. Or is calcification so small a flux that it does not impact carbonate chemistry on the coasts? This seems hard to believe...

Line 204: Say 25C, for consistency.

Line 255-256: Also along the east coast, the Gulf of Maine and northwards...

Line 370-372: Perhaps a call back to fig. 1 here also?

Line 422: Is it presumptuous to call out in particular Northern areas like the Gulf of Alaska and the Gulf of Maine, with high RF values, as places that are particularly sensitive? Might also be a help to those areas policy-wise if the press gets involved.

FIGURES

Figure 1: Panel d: open system is a different shade of green from the rest.

Figure 5: what is the arrow doing in panel g?

SUPPLEMENTAL

I really like Figure S9.

Adam V. Subhas

Reviewer #2 (Remarks to the Author):

I thank the authors for their thoughtful responses to the critiques. I believe they have done an adequate job of responding to all the major critiques. My only remaining criticism is that I still think the intro is too dense for a broad audience. I understand why they want to include information on first principles, but I think a lot of that information, starting at line 89, would be better suited for either the supplement or the methods section and replaced with text on the broader importance of understanding broad scale patterns in carbonate chemistry.

Reviewer #3 (Remarks to the Author):

Dear Authors,

I read the first version of this manuscript in August 2019 during a cruise... it took me a while (a lot!) to understand and check the calculations and general message of the results.

I have read it with "new eyes" in this new 2020 year and I am very happy to feel very satisfied with both the answers to me and the other referees, and also very happy to check the great improvement of the ms, methods, organization, results description and "discussion".

I just hope the other referees feel the same and accept the publication of this manuscript.

I would like just to state .. that I really envy the OAP in the States... my organization IEO is right now bankrupted, any monitoring activity has been a nightmare for the last 4 years and right now stopped with poor hope of any restoration.

So keep measuring (also pH!!!) and sharing your knowledge and data.

Well done!

Best regards

Marta

Point-by-point Responses (3/15)

REVIEWERS' COMMENTS:

Reviewer #1 (Remarks to the Author):

This review is for the manuscript entitled "Controls on Surface Water Carbonate Chemistry along North American Ocean Margins" by Wei-Jun Cai et al., which I reviewed in its previous form. I am pleased to say that the manuscript has been much improved, and I recommend it for publication with minor revisions. The main text is more streamlined, and the discussion of timescales is very nice and indeed shores up the authors' conclusions, from seasonal timescales on upwards. I include the comments below, most of which I believe the authors could take or leave. However, at the end, I think it may be worth mentioning the Revelle Factor analysis more strongly in the conclusions. This may help to drive the point home that northern-latitude coastal ecosystems, both in the Pacific and Atlantic sectors, are particularly vulnerable to anthropogenic CO₂ forcings.

Response: We appreciate the reviewer's previous suggestions which have greatly improved the quality of this paper. The few suggestions made in this round are also very good and are adopted. In particular, we briefly mentioned the possibility of the CaCO₃ precipitation as a reason for high pCO₂ and low pH in southern Gulf of Mexico. We also bring home the implications of Revelle Factor distribution in the concluding lines of the paper as the reviewer suggested. Thank you so very much!

Detailed comments:

TEXT

Abstract:

Lines 49-51: this is good.

Introduction:

Line 72: I would put a sentence here that states this section's aims, i.e. constructing a theoretical framework to understand temperature effects on the carbonate system. Right now I feel like the point is a little buried.

Response: Good suggestion. However, if we do this here (now line 70), we would have to skip the next paragraph and go into establishing the framework right away. Instead, we take the suggestion and add a phrase in the beginning the next paragraph starting in line 86 (now line 84).

Line 73-85: This paragraph feels a bit lost. I'm not sure it belongs here as it disrupts the temperature discussion. Perhaps it belongs down in the final Discussion, or placed somewhere else in the text?

Response: This 2nd paragraph (now line 71-83) is to introduce the various important processes

in coastal oceans (some are different from those in open oceans). Our logic is to start in the 1st paragraph “what are OAs in open oceans and coastal oceans and what have been done and not done in OA research,” followed by this 2nd paragraph “what are the important processes in coastal oceans affecting carbonate systems” before we go into establishing the theoretical framework of temperature effects of the open and closed systems.

That said we do agree with the reviewer that there is a need to connect smoothly these three parts. We now improved the transitions in the beginning of the 2nd and 3rd paragraphs. Thanks.

Results:

Lines 159-161: I think you need to make the point of doing the 25C calculation much more clear. What do you hope to see by comparing the SST and 25C-normalized plots? What is the usefulness of doing such a comparison? Spelling this out will make this whole section flow better, I think.

Response: We agree and modify the sentences to make that clearer. The new sentences are:

By normalizing the $p\text{CO}_2$ and pH values to 25°C (i.e., setting the thermodynamic constants to 25°C in a closed system calculation), a south-to-north pattern becomes apparent with high $p\text{CO}_{2@25\text{C}}$ and low $\text{pH}_{@25\text{C}}$ values in the cold northern regions. A comparison of $p\text{CO}_2$ and pH distributions at in situ SST and at 25°C not only reveals the important role of temperature in seawater carbonate system thermodynamic equilibrium but, perhaps more importantly, also the role of gas exchange in eliminating the temperature-induced air-sea disequilibrium. In other words, high DIC/TA exists in cold northern waters mainly due to atmospheric CO_2 uptake induced by low sea surface temperature.

Lines 181-183: Have you thought about the role of calcification in modifying Omega (and carbonate ion) along the coasts? I am surprised this doesn't feature at all in the discussion, actually, given the threat to calcifiers of OA. Or is calcification so small a flux that it does not impact carbonate chemistry on the coasts? This seems hard to believe...

Response: The main focus of this paper is to illustrate the large spatial scale distribution patterns as a dynamic balance between atmospheric equilibrium and disturbances by local processes and how the two temperature effects play a particularly important role. Thus, we do not feel that we can afford the space to get into various local details too much (for example, we do not feel we should introduce this process with equations in the Introduction). However, we agree with the reviewer that this issue is important and we shouldn't totally ignore it in such a paper with a perspective or review undertone. Thus we choose to touch upon this subject lightly in southern Gulf of Mexico carbonate bank regions where this is mostly likely to be quite important:

(lines 163-173) Carbonate precipitation is another process that could be contributing to the relatively high $p\text{CO}_2$ and low pH conditions. This is supported by a slightly lower TA/SSS ratio (Fig. 2e) and a higher DIC/TA ratio (Fig. 2f, Supplementary Fig. 4) in southern GOM, indicating more TA removal than DIC removal. Precipitation likely occurs in waters above CaCO_3 -rich

banks at the Florida Keys and Yucatan peninsula, but can also happen in other areas⁴⁵.

Line 204: Say 25C, for consistency.

Response: Okay.

Line 255-256: Also along the east coast, the Gulf of Maine and northwards...

Response: Good point. See modified sentence in now lines 258-259: The cold waters of the northeastern margins, Alaskan shelf, and the CCS upwelling centers are approaching this point and are thus most vulnerable to ocean acidification.

Line 370-372: Perhaps a call back to fig. 1 here also?

Response: Yes, now in line 378.

Line 422: Is it presumptuous to call out in particular Northern areas like the Gulf of Alaska and the Gulf of Maine, with high RF values, as places that are particularly sensitive? Might also be a help to those areas policy-wise if the press gets involved.

Response: We can't agree more with you though we also want to be cautious. Thank you for the suggestion. We added a final sentence in the conclusion: In particular, the Revelle Factor distribution clearly draws our attention to ecosystems—including the CCS and northern-latitude coastal regions, such as the Gulf of Maine (Atlantic) and the Gulf of Alaska (Pacific)—as being particularly sensitive and vulnerable to anthropogenic CO₂ forcing.

FIGURES

Figure 1: Panel d: open system is a different shade of green from the rest.

Response: Fixed. Thanks.

Figure 5: what is the arrow doing in panel g?

Responses: In the color bar we limited SST range to 10-20. But the Mexican Pacific data are of 29C. So we used the arrow to show that. (if we use a larger range of 10-30, most other data cannot be differentiated). Probably the arrow was shifted in the review's online version. We now include the arrow as part of the figure so it won't be shifted.

SUPPLEMENTAL

I really like Figure S9.

Responses: Thanks.

Adam V. Subhas

We appreciate very much your many excellent suggestions in this and the previous reviews

which have greatly improved the quality of this paper.

Reviewer #2 (Remarks to the Author):

I thank the authors for their thoughtful responses to the critiques. I believe they have done an adequate job of responding to all the major critiques. My only remaining criticism is that I still think the intro is too dense for a broad audience. I understand why they want to include information on first principles, but I think a lot of that information, starting at line 89, would be better suited for either the supplement or the methods section and replaced with text on the broader importance of understanding broad scale patterns in carbonate chemistry.

Responses: We appreciate very much your many excellent suggestions in this and the previous reviews which have greatly improved the quality of this paper.

Reviewer #3 (Remarks to the Author):

Dear Authors,

I read the first version of this manuscript in August 2019 during a cruise... it took me a while (a lot!) to understand and check the calculations and general message of the results.

I have read it with "new eyes" in this new 2020 year and I am very happy to feel very satisfied with both the answers to me and the other referees, and also very happy to check the great improvement of the ms, methods, organization, results description and "discussion".

I just hope the other referees feel the same and accept the publication of this manuscript.

I would like just to state .. that I really envy the OAP in the States... my organization IEO is right now bankrupted, any monitoring activity has been a nightmare for the last 4 years and right now stopped with poor hope of any restoration.

So keep measuring (also pH!!!) and sharing your knowledge and data.

Well done!

Best regards

Marta

Responses: We appreciate very much your many excellent suggestions in this and the previous reviews which have greatly improved the quality of this paper.

PD:... please check in the text the references to the Supplementary material .. several of them are mixed.

Responses: Thanks for alerting us. Sorry we had the main text and the Methods used two different references and forgot to merge them. We now corrected this.